# Low-dose decitabine priming endows CAR T cells with enhanced and persistent antitumour potential via epigenetic reprogramming

Yao Wang[1,2,4], Chuan Tong[1,4], Hanren Dai[1,4], Zhiqiang Wu[1,4], Xiao Han[1,4], Yelei Guo[1], Deyun Chen[1], Jianshu Wei[1], Dongdong Ti[1], Zongzhi Liu[3], Qian Mei[1], Xiang Li[1], Liang Dong[1], Jing Nie[1], Yajing Zhang[2] & Weidong Han ⬥ [1,2✉]

Insufficient eradication capacity and dysfunction are common occurrences in T cells that characterize cancer immunotherapy failure. De novo DNA methylation promotes T cell exhaustion, whereas methylation inhibition enhances T cell rejuvenation in vivo. Decitabine, a DNA methyltransferase inhibitor approved for clinical use, may provide a means of modifying exhaustion-associated DNA methylation programmes. Herein, anti-tumour activities, cytokine production, and proliferation are enhanced in decitabine-treated chimeric antigen receptor T (dCAR T) cells both in vitro and in vivo. Additionally, dCAR T cells can eradicate bulky tumours at a low-dose and establish effective recall responses upon tumour rechallenge. Antigen-expressing tumour cells trigger higher expression levels of memory-, proliferation- and cytokine production-associated genes in dCAR T cells. Tumour-infiltrating dCAR T cells retain a relatively high expression of memory-related genes and low expression of exhaustion-related genes in vivo. In vitro administration of decitabine may represent an option for the generation of CAR T cells with improved anti-tumour properties.

[1] Department of Molecular Biology and Immunology, Institute of Basic Medicine, Chinese People's Liberation Army General Hospital, Beijing, China. [2] Department of Bio-therapeutic, the First Medical Centre, Chinese People's Liberation Army General Hospital, Beijing, China. [3] Key Laboratory of Genomic and Precision Medicine, Beijing Institute of Genomics, Chinese Academy of Sciences, Beijing, China. [4] These authors contributed equally: Yao Wang, Chuan Tong, Hanren Dai, Zhiqiang Wu, Xiao Han. ✉email: hanwdrsw69@yahoo.com

C himeric antigen receptor T-cell (CAR T) immunotherapy targeting the CD19 antigen has shown clinical efficacy in patients with leukaemia and lymphoma[1–5]. However, to date, this treatment has been much less effective for lymphoma than acute lymphoblastic leukaemia (ALL), in part because CAR T cells enter an exhausted state[6–9] characterised by upregulation of inhibitory receptors and loss of effector function. Recently, de novo DNA methylation was shown to promote T-cell exhaustion and limit anti-PD1 immunotherapy, and the inhibition of methylation inhibition enhanced the PD1 blockade-mediated T-cell rejuvenation[10,11]. Moreover, recent studies have indicated that chromatin accessibility, which is accompanied by functional DNA demethylation, is reduced at transcription start site regions in exhausted T cells[12–14]. Relapse of ALL is often associated with limited CAR T-cell persistence, which suggests a loss of active CAR T-cell-mediated surveillance of leukaemia[12,15,16]. The determinants of CAR T-cell persistence, in addition to inherent T-cell quality and initial T-cell phenotype, have not been fully elucidated[12,17]. Preclinical models have shown that long-persisting CAR T cells possessing a more naive/stem/central memory-like T-cell (Tscm/Tcm) phenotype could achieve optimal control of both haematologic and solid tumours[18–20]. Many genes involved in immune function are regulated by CpG methylation[21,22], indicating a role for this process in T-cell function and differentiation[2,12,23,24].

In this work, we show that CAR T cells treated with low-dose decitabine (5-aza-2'-deoxycytidine, DAC) have stronger anti-tumour, proliferation and cytokine release capacities. Specifically, the DNA-reprogramming effects of CAR T cells are triggered by antigen expression after DAC treatment, and CAR T cells treated with DAC still maintain higher memory-associated and relatively lower exhaustion-associated gene expression under tumour cell stimulation in vivo and in vitro. The use of methylation inhibitor is demonstrated to improve the exhaustion of CAR T cells and promote the maintenance of memory phenotype and effector function.

## Results

**Phenotypic characterisation of CAR T cells treated with decitabine.** CAR (CAR-CD19-expressing) T cells were successfully prepared by transducing human peripheral blood mononuclear cells (PBMCs) with a lentivirus encoding an anti-CD19 scFv and 4-1BB/CD3ζ CAR. Very low-dose DAC in the nanomolar range (10 nM to 1000 nM) was added to the CAR T-cell culture on day 3 (Fig. 1a). The addition of DAC at a dose of 10 nM during CAR T-cell culture had little or no effect on CAR T-cell viability or proliferative capacity (Fig. 1b). The mean fluorescence intensity (MFI) of CAR staining in the total CD3-positive CAR T cells was similar in terms of the presence and absence of DAC (Fig. 1c). We observed an increased CD4:CD8 T-cell ratio and an elevation in the central memory (Tcm, CD45RO + CD62L+) and CD25-positive populations among cultured CAR T cells treated with 10 nM DAC (dCAR T cells) compared to CAR T cells after 10 days of culture (Fig. 1d). The populations of regulatory T cells (Tregs) and Th17 cells were very low and did not differ between the dCAR T and CAR T cells (Supplementary Fig. 1). Without target cell stimulation, the dCAR T cells secreted higher levels of the T-cell proliferation-related cytokine IL-2 than the CAR T cells and similar levels of tumour necrosis factor alpha (TNF-α) and interferon-γ (IFN-γ) in (Fig. 1e). DAC suppresses DNA methyltransferases (DNMTs)[25,26]; DNMT3a mainly functions as a de novo methyltransferase to establish DNA methylation and can also methylate hemi-methylated DNA[27]. We found that low-dose, short-term DAC treatment in vitro persistently induced the degradation of DNMT3a (Fig. 1f).

**Transcriptional and epigenetic changes in dCAR T cells.** To further identify different phenotypic and functional patterns and methylation modification patterns, we performed genome-wide transcriptional profiling and chip-based 850k whole-DNA methylome analysis of the dCAR T and CAR T cells in a "resting state" (not stimulated by antigens or tumour cells). The average beta values of the whole-DNA methylome were reduced in the dCAR T cells compared to the CAR T cells (Fig. 2a). In total, 12809 CpG sites exhibited differential methylation (Fig. 2b), and 1034 gene promoter-associated CpG sites were downregulated in the dCAR T cells (Supplementary Data 1). Gene ontology (GO) analysis showed that the differentially methylated CpG site-associated genes were enriched in T-cell differentiation, cell death and T-cell differentiation and ageing (Fig. 2c). Similar to the results of the methylation analysis, although to a less extent, the results of transcriptional profiling showed that the dCAR T cells exhibited characteristic transcriptional profiles that were not quite identical to those of the CAR T cells (Fig. 2c, d). Gene set enrichment analysis (GSEA) revealed upregulation of memory- and proliferation-associated genes; downregulation of T-cell inhibitor-, exhaustion/activation- and cell death-associated genes (Fig. 2e, f), such as transcription factor 7 (TCF7), B-cell lymphoma 6 protein (BCL6), lymphoid enhancer-binding factor 1 (LEF1) and Interleukin 7 Receptor (IL7R)[23,28]; and enhanced downregulation of T-cell exhaustion/death- and inhibitor-related genes, such as eomesodermin (EOMES), lymphocyte-activation gene 3 (LAG3), cytotoxic T lymphocyte-associated protein 4 (CTLA-4) and nuclear receptor subfamily 4 group A member 3 (NR4A3) in the dCAR T cells compared to the CAR T cells[29–31] (Fig. 2g). A previous report showed that DAC can promote T-cell proliferation in vivo[32]. Interestingly, although T-cell activation- and differentiation-related factors were relatively downregulated, the dCAR T cells still upregulated expression of proliferation- and memory-associated genes (Fig. 2e, g). This result was also observed in the expanded T-cell samples, although to a lesser extent, as the transcription profile results of five donors revealed the upregulated expression of memory- and proliferation-associated genes and enhanced downregulation of the expression of T-cell inhibitor-, death- and activity/exhaustion-related genes in the dCAR T cells compared to the CAR T cells (Supplementary Fig. 2). The differentially expressed genes in the transcription profiles also had coordinately altered methylation sites, especially memory-associated genes, such as BCL6, TCF7, LEF1 and IL7R, and exhaustion/ageing-associated genes, such as AKT3, EOMES, NR4A3 and CTLA-4 (Fig. 2h). Collectively, these results showed that compared with the CAR T cells, the dCAR T cells displayed a different gene expression and DNA epigenetic statuses.

**The rapid and enhanced killing ability of dCAR T cells.** Next, we established an electrical impedance-based tumour cell culture system (xCELLigence) to compare the duration of the antitumour activities of dCAR and CAR T cells. Compared to the CAR T cells, the dCAR T cells were more effective at killing Raji tumour cells (CD19- and CD20-positive) over 140 h (Fig. 3a). Furthermore, upon a co-culture at an effector-to-target (E:T) ratio of 1:30 for 64 h, the dCAR T cells were significantly better at eliminating Raji tumour cells than the CAR T cells (Fig. 3b, c and Supplementary Movies 1–2). Notably, many of the data points were acquired below an E:T ratio of 1:1, demonstrating that the activity of the dCAR T cells was strong and rapid (Fig. 3c). This conclusion was confirmed by the analysis of CD107a expression in dCAR T cells. After 0.5 h and 1 h of co-culture with Raji cells, CD107a expression on the dCAR T-cell surface was higher than that on the CAR T-cell surface (Fig. 3d). Upregulated CD107a

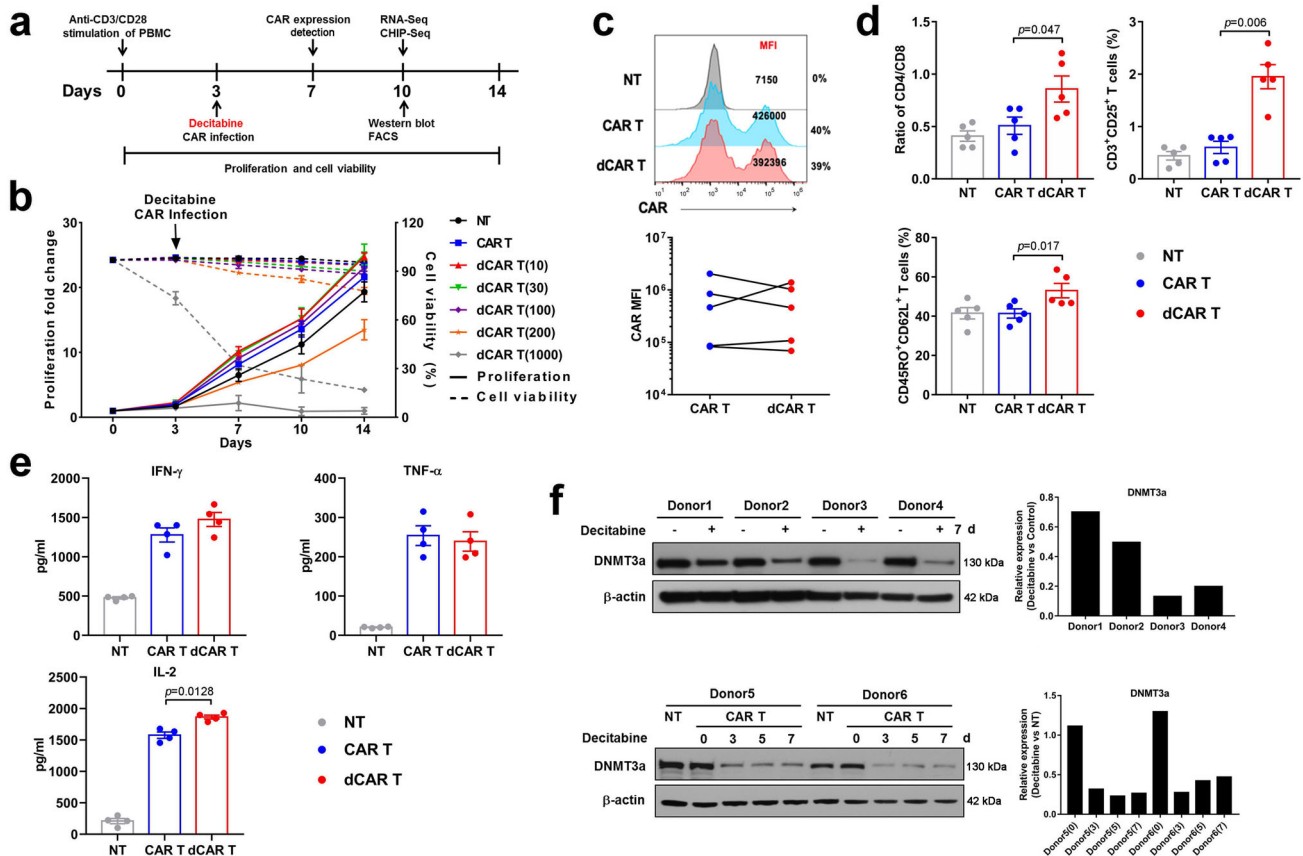

**Fig. 1 CAR T cells treated with decitabine (DAC) show less differentiation and enhanced memory. a** Flowchart of dCAR T cell (CAR T cells with decitabine treatment) culture and functional tests. DAC was added at the same time as the CAR gene infection and removed after 24 h. **b** Dose-dependent effect of DAC in CAR T-cell culture ($n = 3$ samples examined over three independent experiments). NT: un-transduced T cells. The number in brackets represents the concentration of DAC, nM. **c** Upper panel: Representative flow cytometric profile of CAR-positive T cells (percentages) and mean fluorescence intensity (MFI) indicating CAR expression. Lower panel: MFI of CAR on CAR T and dCAR T cells ($n = 5$ samples examined over three independent experiments). **d** Histogram plots of CD3, CD4, CD8, CD62L, CD45RO and CD25 expression on NT, CAR T and dCAR T cells after 10 days of cell culture ($n = 5$ samples examined over three independent experiments). **e** Equal numbers of NT, CAR T and dCAR T cells were collected on day 7 and cultured for 24 h with new medium without human IL-2. Then, cytokine production by NT, CAR T and dCAR T cells secreted was measured by ELISA ($n = 4$ samples examined over three independent experiments). **f** Western blot analysis of DNMT3a levels in CAR T cells treated with or without DAC (upper panel, $n = 4$; lower panel, $n = 2$). Control: CAR T cells without DAC treatment. The data (**d, e**) are presented as the mean ± s.e.m. Two-tailed paired $t$ tests were used for statistical analysis.

expression indicates degranulation and correlates with cytotoxicity[33]. We analysed the concentrations of cytokines and chemokines in the supernatants of dCAR and CAR T cells co-cultured with Raji cells. The dCAR T cells secreted significantly more cytokines and chemokines than the CAR T cells (Fig. 3e). To examine whether these effects were coupled with T-cell proliferation, we stained dCAR and CAR T cells for Ki67 and HLA-DR upon antigen stimulation. The dCAR T cells had prominently higher levels of Ki67 and HLA-DR than the CAR T cells after 24 h of co-culture with Raji tumour cells (Fig. 3f). In addition, tandem CAR T cells targeting both CD19 and CD20 also showed increased cytotoxicity, cytokine production and proliferation in vitro after treatment with 10 nM DAC (Supplementary Fig. 3). We next investigated whether the proliferative capability and exhaustion profile of the dCAR T cells in response to repeated antigen stimulation in vitro differed from those of the CAR T cells (Fig. 3g). Higher proliferation rates upon the second and third rounds of antigen stimulation than the CAR T cells were consistently observed in different donors (Fig. 3h). Furthermore, downregulation of the expression of CD3 + PD1+ and CD3 + EOMES + expression and upregulation of the expression of CD3 + CD25+ and Tcm marker expression were observed with

repeated antigen stimulation in dCAR T cells from different donors subjected to repeated antigen stimulation (Fig. 3i). These results were confirmed by further extension of the tumour cell stimulation time. After 2 weeks of constant stimulation by low-dose Raji cells, CAR T cells exhibited higher proportions of PD1 + cells than dCAR T cells did (Supplementary Fig. 4a). Compared to CAR T cells, dCAR T cells produced higher levels of IL-2, TNF-α and IFN-γ upon restimulation with Raji cells (Supplementary Fig. 4b). These results indicated that the dCAR T cells retained robust effector functions and exhibited upregulated expression of memory-associated markers. The data obtained following repeated antigen stimulation suggested that prevention of exhaustion could be one mechanism by which DAC treatment enhanced antitumour functionality.

**Improved cell function in both CD4 and CD8 dCAR T cells.** After observing the enhanced cell function of the dCAR T cells in vitro, we further explored the effect of DAC on CD4-positive and CD8-positive cells which were purified from the initial PMBCs (Fig. 4a). Treatment with DAC did not change the MFI or the efficiency of CAR expression on the cell surface (Fig. 4b).

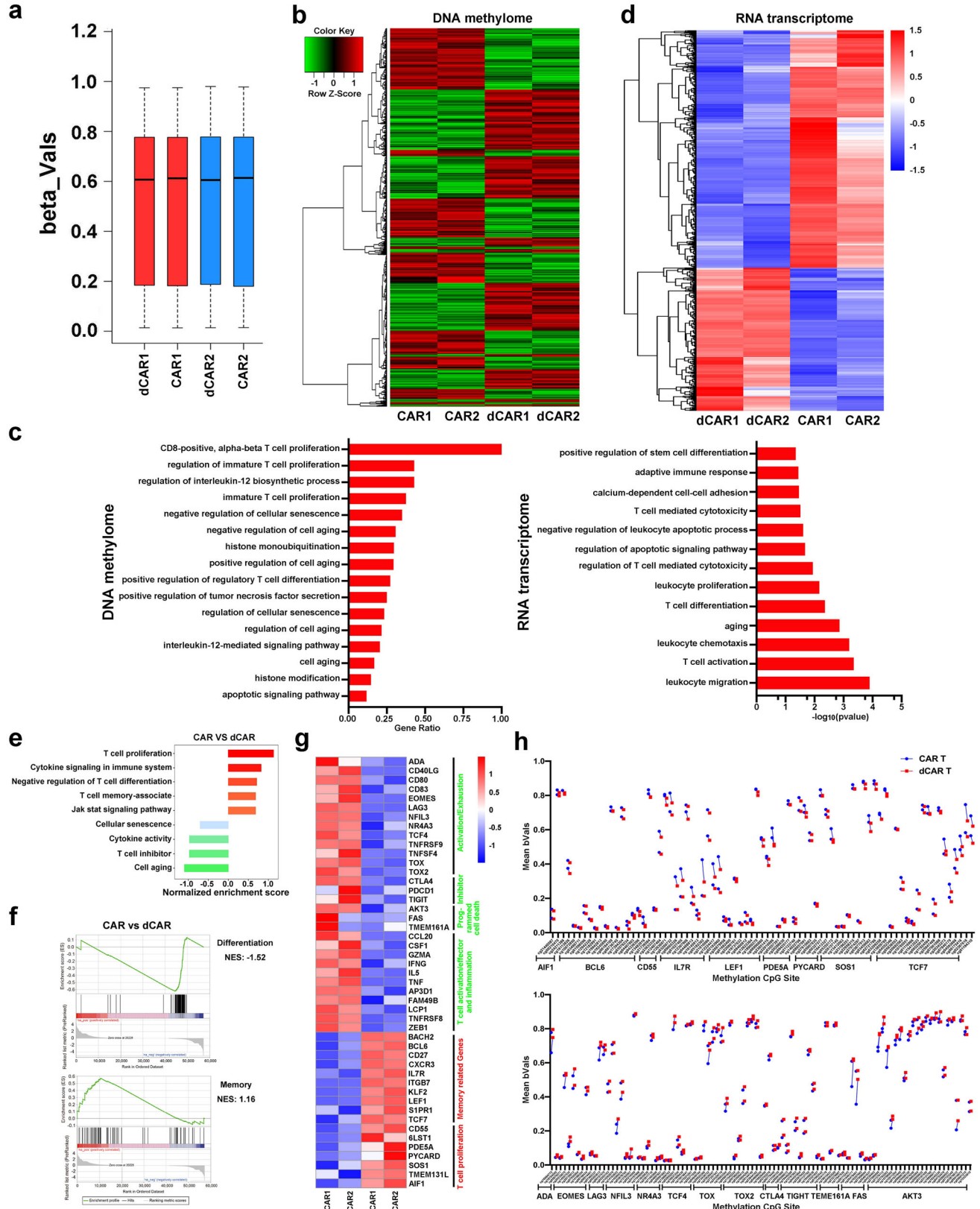

Notably, the expression of CD62L was more concentrated on CD4-positive CAR T cells after DAC treatment (Fig. 4c). Interestingly, the proliferation of CD4-positive cells was significantly increased, while the growth rate of CD8-positive T cells was decreased after DAC treatment (Fig. 4d), which may explain the increase in the CD4:CD8 cell ratio in dCAR T cells (Fig. 1d).

Furthermore, the amplification capacity of both CD4-positive and CD8-positive T cells was higher in the dCAR T-cell group than that of in the CAR T-cell group under repeated antigen stimulation (Fig. 4e). In addition, both CD4-positive and CD8-positive dCAR T cells presented the improved killing ability to Raji cells at an E:T ratio of 1:5 (Fig. 4f). After co-culture with tumour cells for

**Fig. 2 Transcriptional signatures and DNA methylation characteristics of CAR T cells after DAC treatment.** Gene expression and DNA methylome profiles of CAR and dCAR T cells after 7 days of CAR gene infection. **a** The methylation status of all the probes was denoted as beta (β) value, which is the ratio of the methylated probe intensity to the overall probe intensity. The mean beta values of CAR T and dCAR T cells measured by chip-based 850k whole-DNA methylome analysis ($n = 2$) are shown. **b** Hierarchical clustering analysis of significant differential methylation CpG sites among dCAR T cells compared to CAR T cells ($n = 2$). **c** Left: GO functional clustering of significantly differentially expressed CpG site-related genes were enriched in biological processes in dCAR T cells; data are presented as the ratio of the number of CpG-related genes to the number of GO item genes ($n = 2$). Right: GO functional clustering of genes that were upregulated for biological processes in dCAR T cells ($n = 2$). **d** Hierarchical clustering of the RNA-seq analysis results shows differentially expressed genes between dCAR T cells and CAR T cells ($n = 2$). **e** Normalised enrichment scores of significantly up- or downregulated gene sets in the CAR T cell versus dCAR T-cell groups ($n = 2$), as determined by GSEA using the MSigDB C7 gene ontology sets. **f** Representative GSEA enrichment plot demonstrating the upregulation of memory-related genes and downregulation of differentiation-, exhaustion- and programming death-related genes in dCAR T cells versus CAR T cells ($n = 2$). **g** Differentially expressed genes between CAR and dCAR T cells and heatmap demonstrating the different expression profiles of the T-cell function genes for CAR T cells ($n = 2$). **h** Mean beta values for the differential CpG sites in the specified genes ($n = 2$). Significantly differentially CpG sites in (**b**) and (**h**) were calculated by generalised linear models (v3.36.2) ($P$ value <0.05, fold change (log2 scale) ≥1 or ≤−1). Significantly differentially expressed genes in (**d**) and (**g**) were calculated the Wald test (as implemented in DESeq2) ($P$ value <0.01, fold change (log2 scale) ≥1 or ≤−1).

24 h, HLA-DR expression was higher increased in both CD4- and CD8-positive dCAR T cells (Fig. 4g). In addition, both the CD4-positive and CD8-positive dCAR T cells upregulated CD107a degranulation when co-cultured with Raji cells for 0.5 h (Fig. 4h). The CD4-positive dCAR T cells showed increased cytokine and chemokine secretion (Fig. 4i).

**Genomic basis for enhanced dCAR T-cell reactivity.** Recent studies have shown that the development of memory characteristics is associated with increased antitumour efficacy and persistence in adoptively transferred T-cell subsets[34,35]. In addition, the interaction of tumour cells with CAR T cells can induce the expression of immunosuppressive molecules, such as *CTLA-4*, *EOMES* and *PD1*, leading to T-cell exhaustion or dysfunction[7]. To further verify the molecular mechanisms underlying the improved antitumour function of the dCAR T cells under antigenic stimulation, we analysed the transcriptional profiles of CAR T cells after repeated co-culture with tumour cells in vitro. As expected, the dCAR T cells and CAR T cells showed differences in their methylation and transcription profiles after antigen stimulation in vitro (Fig. 5a, b). After tumour cell stimulation, the difference in methylated CpG sites between the dCAR T cells and the CAR T cells was not the same as that in cells in the "resting state". A total of 25631 CpG sites, including 6005 promoter-related CpG sites (Supplementary Data. 2), were differentially altered in the dCAR T-cell group ($P < 0.05$). GO analysis showed that these differentially methylated CpG sites were enriched in genes related to T-cell costimulation, cell death, the WNT signalling pathway and T-cell receptor signalling pathway (Fig. 5c). The results of GO analysis showed that differentially altered genes in the dCAR T cells were enriched in cytokine production/secretion, T-cell migration and T-cell proliferation (Fig. 5c). GSEA revealed decreased expression of exhaustion-associated and inhibitor-related genes and enhanced enrichment expression of memory-associated and cell proliferation-related genes in the dCAR T cells compared with the CAR T cells (Fig. 5d). Consistent with those before antigen stimulation, the dCAR T cells after antigen stimulation retained the relatively upregulated expression of memory-associated transcription factors, including *TCF7*, *BCL6* and *LEF1*, as well as memory-related genes, such as *CCR7*, *IL7R* and *POU6F1*. Unexpectedly, *CTLA-4*, *CD38*, *EOMES* and *CD244* expression was downregulated in the dCAR T cells (Fig. 5e). In addition, although cell proliferation-associated genes, including *SELL*, *RPS3*, *CD55*, and *SOS1*, were upregulated in the dCAR T cells compared with the CAR T cells after antigen stimulation, exhaustion- and programmed death-related genes, including *AKT3*, *BAK1*, *DDIAS*, *HTRA2* and *RRN3*, did not show upregulated expression, as observed with T-cell proliferation;

instead, these genes were downregulated after different treatments (Fig. 5e). These results were also observed in extended biological RNA-seq (Supplementary Fig. 5). After tumour cell stimulation, the corresponding methylated CpG sites in the differentially expressed genes of the dCAR T cells also changed in a synchronous manner. These genes included memory-related genes such as *ADCY1*, *BCL6*, *CCR7*, *IL7R*, *LEF1* and *TCF7* and inhibitor- or cell death-related genes such as *AKT3*, *CASP3*, *EOMES* and *TOX2* (Fig. 5f). Importantly, regardless of whether the CAR T cells or the dCAR T cells were examined, their DNA methylation was significantly altered by antigen exposure (Supplementary Fig. 6a). We performed Venn diagram overlay analysis of the two CAR T-cell groups for genes harbouring promoter-associated CpG sites before and after antigen stimulation, and the data showed that more than 6000 gene promoter-associated CpG sites were affected by DAC treatment when the cells were compared before and after antigen exposure (Supplementary Fig. 6b). Some specific gene promoter-associated CpG sites among these sites are consistent with the regulation of changes in the transcriptome, including those in *TCF7*, *BAD*, *BCL6* and other methylation and acetylation-related genes (Supplementary Fig. 6c). These CpG sites may regulate the biological effects of the dCAR T cells, but this requires further validation. Our results suggest that the reprogramming effect on dCAR T cells is exerted and magnified after antigen exposure through treatment with DAC rather than by CAR T-cell intrinsic factors (Fig. 5g).

**Enhanced in vivo antitumour activity of dCAR T cells.** The above findings were subsequently evaluated in vivo, as the dCAR T cells exhibited enhanced cytotoxicity and proliferation and reduced exhaustion. First, we utilized CAR-19-expressing T cells as a reference to assess the cytolytic potential and persistence of dCAR T cells in NPG (NOD-Prkdc$^{scid}$Il2r$^{gnull}$/Vst) mice in vivo with an established Raji cell-based non-Hodgkin's lymphoma (NHL) model (Fig. 6a). Treatment with an established CAR T-cell dose ($1 \times 10^7$ cells) effectively controlled Raji cells[36]. Both dCAR T and CAR T-cell treatment with $1 \times 10^7$ CAR T cells effectively controlled tumour growth. Compared to the CAR T cells, the dCAR T cells showed a slight shortened change in tumour control time, but this difference was not significant (Fig. 6b and Supplementary Fig. 7). Similar results were also observed with tandem CAR T cells (Fig. 6c and Supplementary Fig. 7). However, CAR gene numbers in tumours (3 days after infusion) and peripheral blood (PB) were substantially higher in the dCAR T-cell group (Fig. 6d). Importantly, the dCAR T cells could elicit effective recall responses (Fig. 6b, e), achieving T-cell reamplification and complete tumour control following tumour rechallenge (Fig. 6e, f); in contrast, the CAR T cells could not

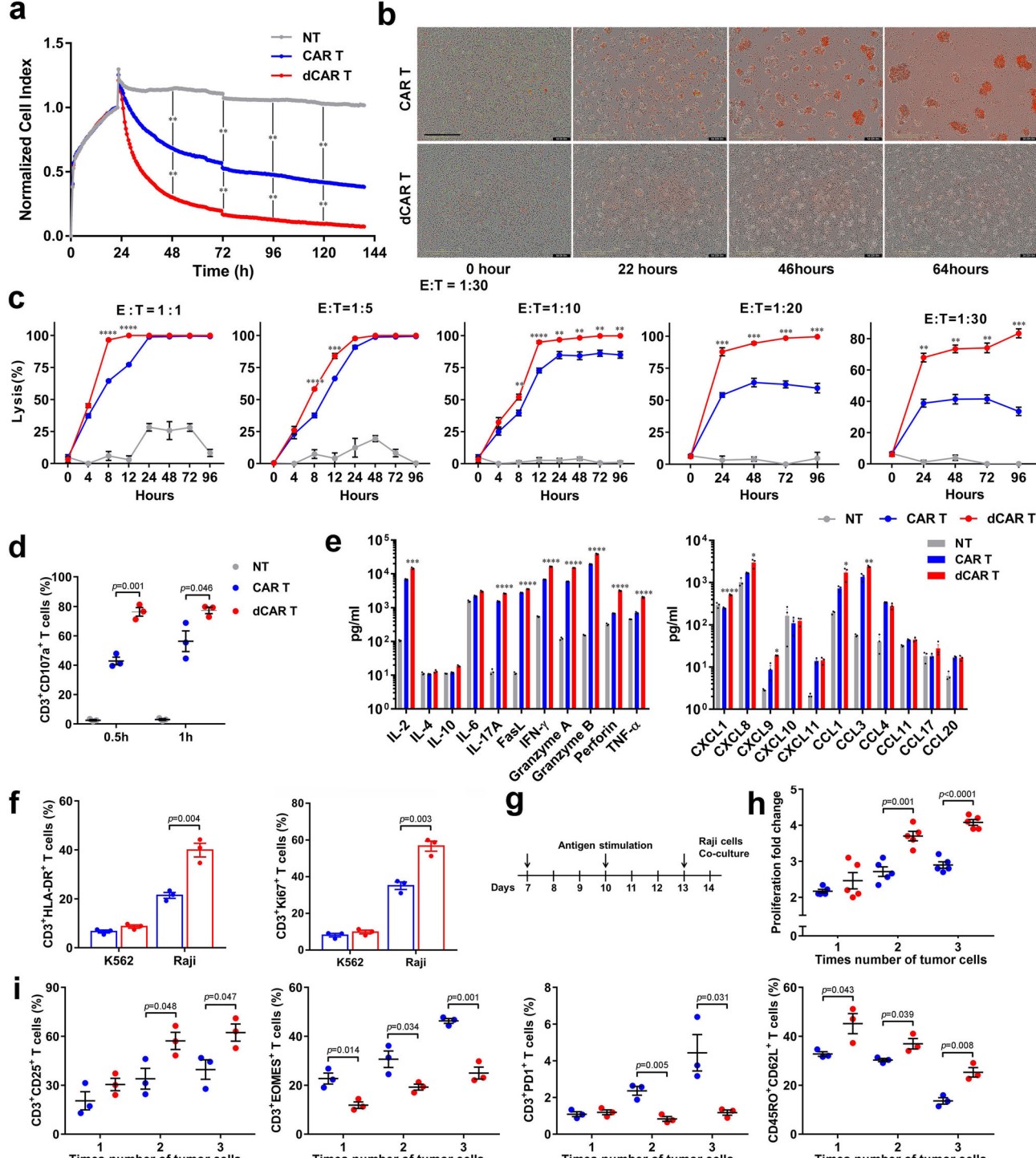

**Fig. 3 dCAR T cells exhibit enhanced antitumour reactivity and the maintenance of a memory-like phenotype at a low effector:target ratio.**
**a** Continuous graphical output of cell index values determined using the xCELLigence Impedance system from NT, CAR T and dCAR T cells co-cultured with Raji cells at an effector-to-target (E:T) ratio of 1:5 up to the 140-h time point. The density of Raji cell changes from baseline at each time point was calculated and compared using two-tailed paired $t$ tests. **$P < 0.01$. ($n = 2$). **b** Representative phase: dCAR T or CAR T cells co-cultured with Raji cells at an E:T ratio of 1:30 ($n = 2$). Green fluorescent (nuclear-restricted RFP) cells were CAR T or dCAR T cells. Red fluorescent (Yoyo3) cells were counted as dead cells. Scale bar: 400 μm. **c** Cytotoxic analysis of dCAR and CAR T cells co-cultured with Raji cells at an E:T ratio of 1:1 to 1:30 (1:1, 1:5, 1: 10 ($n = 4$); E:T = 1:20 and 1:30 ($n = 3$)). **d** CD107a expression on CAR T cells after co-culture with Raji cells at an E:T ratio of 1:1 for 0.5 or 1 h ($n = 3$). **e** Cytokine production by dCAR and CAR T cells co-cultured with Raji cells at an E:T ratio of 1:1 for 24 h ($n = 4$). **f** Histogram plots of HLA-DR and Ki67 expression on of CAR and dCAR T cells after co-culture with Raji cells at an E:T ratio of 1:1 for 24 h ($n = 3$). **g** Pattern of repeated antigen stimulation in vitro. ("↓" represents the time point when Raji cells were added). **h** Cell count analysis of CAR T-cell proliferation as measured after 24 h of co-culture with Raji cells at every time point ($n = 5$). **i** Continuous testing of CD3/CD25, CD3/EOMES, CD3/PD1 and CD45RO/CD62L expression on CAR and dCAR T cells was performed after 24 h of co-culture with Raji cells at an E:T ratio of 1:1 ($n = 3$). Data for all panels are presented as the mean ± s.e.m. $P$ values for **c**–**f**, **h** and **i** were calculated by two-tailed unpaired $t$ tests. *$P < 0.05$, **$P < 0.01$, ***$P < 0.001$ and ****$P < 0.0001$. For **c** and **e**, exact $P$ values are available in the Source Data File. Source data are provided as a Source Data file.

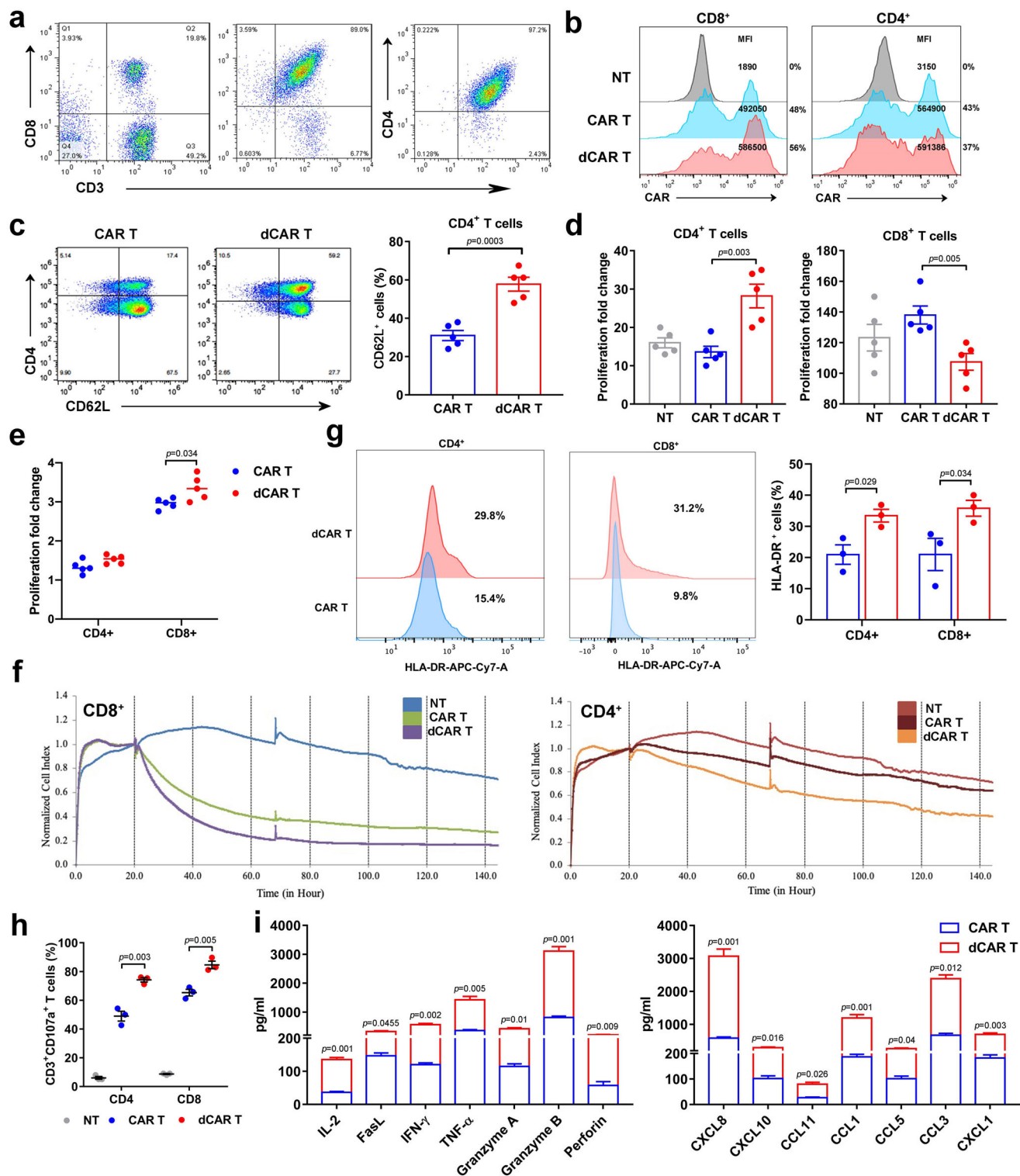

control tumour rechallenge and were no longer detectable in vivo (Fig. 6e).

**Advantages of dCAR T cells in terms of antitumour ability.** Then, we evaluated the tumour burdens through a devised "stress test" based on the previous studies[19,37,38], in which the CAR T-cell dosage was gradually lowered to reveal the functional limits. Here, we extended the tumour growth time to 3 weeks and different treatment doses of $1 \times 10^7$, $1 \times 10^6$ and $2 \times 10^5$ CAR T cells

were tested (Fig. 7a, f). Under the increasing tumour burden, compared with the CAR T cells, the dCAR T cells resulted in quick elimination of tumour burdens and enhanced survival (Fig. 7b, c and Supplementary Fig. 8), and differences in tumour control were apparent as early as day 14 after T-cell infusion (Fig. 7d). Consistent with the observed rapid tumour clearance, the dCAR T cells showed strong, quick and persistent amplification. The peak of CAR T-cell expansion reached its maximum at 14 days after cell infusion, and this high level was maintained

**Fig. 4 Enhancement of cell function in CD4- and CD8-positive CAR T cells after DAC treatment. a** CD4- or CD8-positive subsets separated at the beginning of CAR T-cell culture underwent magnetic selection processes using human CD8 + or CD4 + T-cell isolation kits (Miltenyi Biotec), respectively, and the negative fraction was collected. **b** Representative percentage of CAR-positive T cells and MFI indicating CAR expression on CD4-positive or CD8-positive CAR T cells. **c** The CD62L expression of CD4-positive T cells in CAR T and dCAR T cells. **d** The proliferation fold change was obtained by the number of cell counts on the 10th day after cell culture compared with the number of cells before cell culture. **e** Cell count analysis of CAR T-cell proliferation as measured after 24 h of co-culture with Raji cells at an E:T ratio of 1:1. The data in **c**, **d** and **e** are from three independent experiments with five samples. **f** Continuous graphical output of cell index values up to the 144-h time point from Raji cells during incubation with NT, CD4-positive and CD8-positive CAR and dCAR T cells at an E:T ratio of 1:5 determined using the xCELLigence Impedance system. ($n = 2$). **g** Flow cytometry analysis of HLA-DR expression in CD4-positive and CD8-positive CAR T cells co-cultured with Raji cells at an E:T ratio of 1:1 for 24 h. **h** Flow cytometry analysis of CD107a expression on CD4-positive and CD8-positive CAR T cells after their co-culture with Raji cells at an E:T ratio of 1:1 for 0.5 h. The data in **g** and **h** are from three independent experiments with three samples. **i** Cytokine production by CD4-positive dCAR and CAR T cells co-cultured with Raji cells at an E:T ratio of 1:1 for 24 h was measured by Luminex assay according to the manufacturer's instructions ($n = 3$). All data are the mean ± s.e.m. $P$ values for all panels were calculated by two-tailed unpaired $t$ tests.

for one month (Fig. 7e). Although the efficacy of tumour elimination decreased in the CAR T and dCAR T-cell groups at the $2 \times 10^5$ cell dose, the dCAR T cells showed better survival than the CAR T cells in both the Nalm-6 ALL and the Raji NHL models (Fig. 7g, h). In addition, the dCAR T cells at the lowest dose could still induce complete remissions in a few instances; in contrast, the CAR T cells did not achieve any tumour control in either model (Fig. 7g, h).

**Functional and phenotypic features of dCAR T cells in vivo.** To further explain the enhanced antitumour ability of the dCAR T cells in vivo, we performed a serious test for CAR T cells in the tumour and peripheral blood. Differentiated T cells lose the ability to produce interleukin 2 (IL-2) because they acquire the ability to kill target cells and release large amounts of IFN-γ[39,40]. The cell and CAR gene numbers (Fig. 8a, b) and levels of cytokines related to T-cell proliferation, such as IL-2 and IL-5, or functional cytokines related to T-cell activities, such as IFN-γ, TNF-α and granzyme, were substantially higher in the dCAR T group than in the CAR T group (Fig. 8c). The results from RNA in situ hybridisation (ISH) showed a large number of infiltrating CAR T cells with significantly elevated expression of IFN-γ, granzyme B and FasL in the tumour sites on day 14 after infusion in the dCAR T group (Fig. 8d). These results suggested that dCAR T cells have the better tumour-homing ability or anti-tumour potential. Consistent with these results, the dCAR T cells showed greater enrichment in memory-related genes by expression profiling, while many inhibitions/exhaustion-related genes were relatively downregulated after antigen stimulation. The CAR T cells in bone marrow showed limited evidence of exhaustion four weeks after infusion in the CAR T group (Fig. 8e). In contrast, fewer than 2% of the T cells in the dCAR T-cell group were triple-positive for the markers TIM3 + LAG3 + PD1 +, and these cells also retained a larger proportion of cells with memory-associated CD62L and activation-associated CD25 expression than other cells (Fig. 8f). To obtain nucleotide resolution of DNA methylation, we performed bisulfite sequencing of TIM3 and PD1 on genomic DNA from in vitro-activated primary T cells cultured in the presence and absence of DAC. TIM3 and PD1 were partially methylated in effector cells on day 7. Moreover, treatment of the activated T cells with DAC resulted in the complete demethylation of CpG sites in TIM3 and PD1 (Supplementary Fig. 9).

**Increased cell function of tumour-infiltrating dCAR T cells.** To further assess phenotypic and genome-wide changes associated with antitumour function, we modified the Raji NHL experimental conditions to delay tumour regression and obtained CAR T cells at the tumour site, which were used to perform phenotype testing and RNA-seq (Fig. 9a). The tumour sizes of the dCAR T-

cell group on day 7 after cell infusion were similar to those of the CAR T-cell group but significantly smaller than those of the CAR T-cell group on day 14 (Fig. 9b). Tumour-infiltrating CAR T-cell recovery was higher in the dCAR T-cell group on day 7 and/or day 14 after cell infusion than in the CAR T-cell group (Fig. 9c), confirming that the dCAR T cells had significantly improved expansion and homing ability advantages in vivo. Tcm marker expression was significantly upregulated in the tumour-infiltrating dCAR T cells, including CD8-positive and CD4-positive dCAR T cells, on the 7th and 14th days after cell infusion (Fig. 9d). The number of CD3 + CD25 + cells among tumour-infiltrating dCAR T cells was significantly increased on both the 7th and 14th days after cell infusion (Fig. 9e). Seven days after cell infusion, CD8 + PD1 + TIM3 + cells were highly expressed among both the tumour-infiltrating CAR T and dCAR T cells (Fig. 9f). The CD8-positive tumour-infiltrating dCAR T cells showed a mild but not significant increase in PD1 + TIM3 + expression from day 7 to day 14, but CD8 + PD1 + TIM3 + expression was strongly upregulated in the tumour-infiltrating CAR T cells from day 7 to day 14 and higher than that in the dCAR T cells on day 14 (Fig. 9f).

Others have described highly infiltrated tumours with a distinct population of TIM3-positive cells that resemble phenotypically exhausted CD8 T cells[13]. Similar to a previous report[41,42], the tumour-infiltrating dCAR T and CAR T cells showed the high levels of TIM3 + PD1 + populations on 7th and 14th days after cell infusion were observed (Fig. 9f). We performed RNA-seq analysis of the tumour-infiltrating dCAR T and CAR T cells to further investigate the T-cell function-associated gene status of the dCAR T cells and differences compared with the CAR T cells. Compared with the cells before infusion, both the tumour-infiltrated CAR T cells and dCAR T cells had significantly altered transcriptional expression profiles (Fig. 10a), and the gene expression profiles of the tumour-infiltrating dCAR T and CAR T cells on days 7 and 14 differed (Fig. 10b). GSEA revealed upregulated the expression of memory-associated and cell proliferation-related GO items in the dCAR T cells compared with the CAR T cells on days 7 and 14 (Fig. 10c). Compared with tumour-infiltrating CAR T cells, tumour-infiltrating dCAR T cells showed the upregulated expression of activation/exhaustion-associated genes on day 7 and downregulated expression of these genes on day 14 after cell infusion (Fig. 10d). Throughout the experimental period, the tumour-infiltrating dCAR T cells had higher levels of memory-associated genes (Supplementary Fig. 10), and the dCAR T-cell group exhibited stronger expansion and greater viability of the tumour-infiltrating dCAR T cells, which showed increased populations of Tcm and CD25-positive cells, than those of the CAR T-cell group (Fig. 9e, f); thus, the relatively less exhausted status of the dCAR T cells in the later period may be because a greater proportion of these cells were stemmed cell-

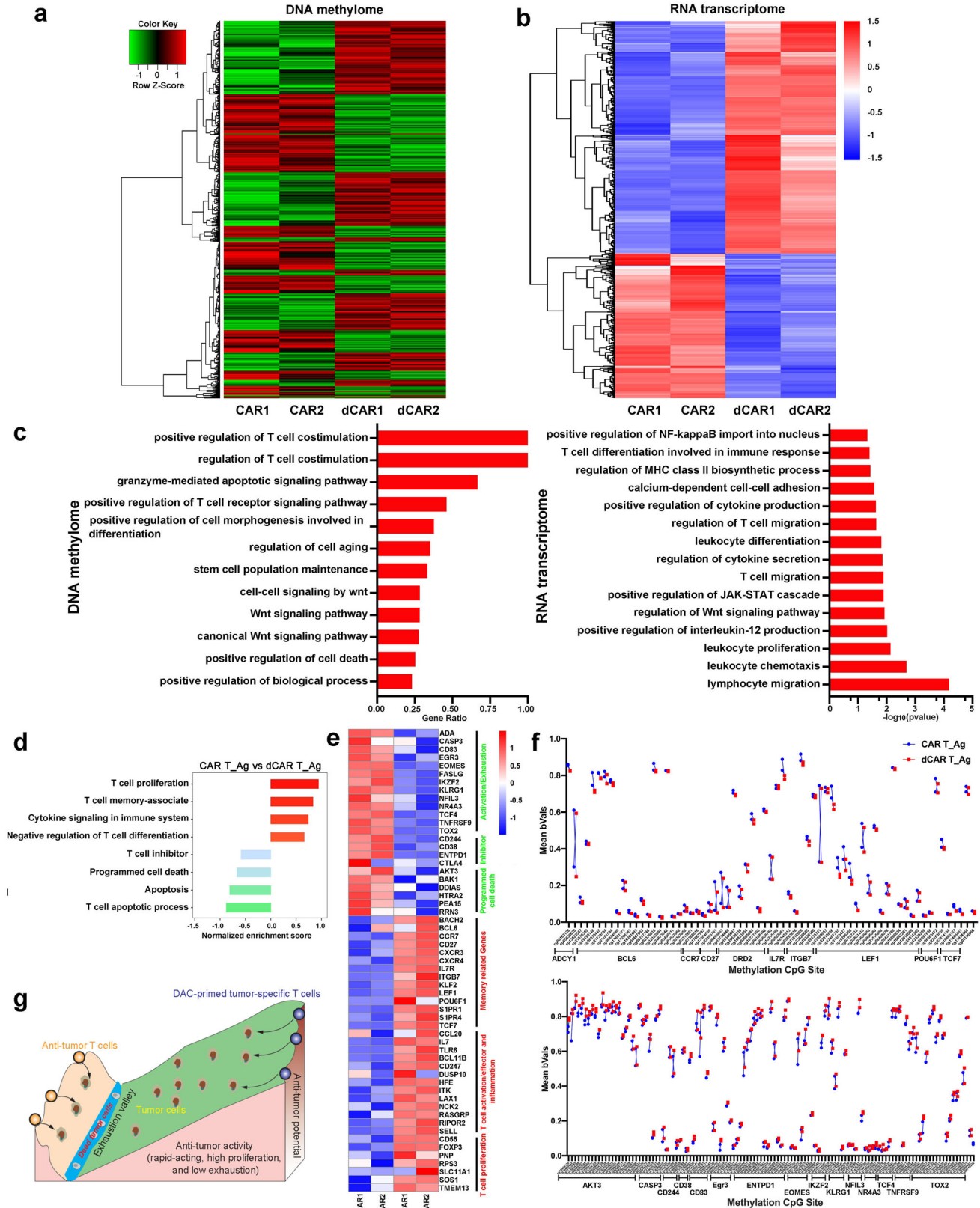

like cells. The tumour-infiltrating dCAR T cells significantly upregulated the expression of cytokine- and T-cell activation-related genes relative to that of the tumour-infiltrating CAR T cells on the 7th day and 14th day (Fig. 10d and Supplementary Fig. 10). Combined with cytokine detection (Fig. 8c, d) and the RNA-seq data, the cytokine-produced function was reduced in the CAR T cells after 14 days of cell infusion; in contrast, the tumour-infiltrating dCAR T cells showed stable in cytokine-produced function. These data suggested that the tumour-infiltrating CAR T cells were in a relatively dysfunctional state relative to the tumour-infiltrating dCAR T cells after cell infusion. To confirm that these gene transcriptome changes were due to

**Fig. 5 The reprogramming effect of DAC on CAR T cells is amplified after antigen stimulation.** All samples were subjected to transcriptome and methylation assays before antigen stimulation and were collected 24 h after antigen stimulation. **a** Hierarchical clustering analysis of significant differential methylation CpG sites among dCAR T cells compared to CAR T cells after antigen stimulation. ($n = 2$). **b** Hierarchical clustering of the RNA-seq analysis results shows differentially expressed genes between dCAR T cells and CAR T cells after co-culture with tumour cells. ($n = 2$). **c** Left panel: GO analysis of significantly differentially expressed CpG site-related gene enrichment in biological processes functions. All data are presented as the ratio of the number of CpG-related genes to the number of pathway genes ($n = 2$). Right panel: GO functional clustering of genes that were upregulated for biological processes in dCAR T cells. **d** Normalised enrichment scores of significantly up- or downregulated gene sets in the CAR T cell versus dCAR T-cell groups ($n = 2$), as determined by GSEA using the MSigDB C7 gene ontology sets. **e** Differentially expressed genes between CAR T and dCAR T cells and a heatmap demonstrating the different expression profiles of the T-cell function-related genes in CAR T cells ($n = 2$). **f** Mean beta values of the differentially expressed CpG sites in specified genes. Upper panel: memory-related and proliferation-related genes. Lower panel: exhaustion-related and inhibitor-related genes. **g** Modified "Waddington valley" diagram showing how DAC reprograms DNA in CAR T cells. This diagram depicting the differential antitumour activity of DAC-treated CAR T cells compared to regular CAR T cells according to results obtained from **a** to **f**. Significantly differentially CpG sites in (**a**) and (**f**) were calculated by generalised linear models (v3.36.2) ($P$ value <0.05, fold change (log2 scale) $\geq 1$ or $\leq -1$). Significantly differentially expressed genes in (**b**) and (**e**) were calculated the Wald test (as implemented in DESeq2) ($P$ value <0.01, fold change (log2 scale) $\geq 1$ or $\leq -1$).

changes in the DAC treatment, we further performed Venn diagram overlay analysis of the dCAR T and CAR T cells before and after infiltration into the tumour site. The dCAR T cells and CAR T cells did not exhibit the same genetic changes (Fig. 10e). These results, combined with DNA methylation results (Supplementary Fig. 6), suggested that the reprogramming effect on dCAR T cells was exerted and magnified after antigen exposure in the tumour site through treatment with DAC rather than by CAR T-cell intrinsic factors.

## Discussion

The application of CAR T-cell therapy has become one of the most promising areas for B-cell malignant haematological tumours. However, the success rate of CAR T cells against chronic lymphocytic leukaemia (CLL)[43] and solid tumours[5,44] is much lower than that against ALL or NHL; even though ALL has the highest success rate, it has an ~40% recurrence rate[45], in part because of the insufficient efficacy and memory persistence[16] of CAR T cells in vivo. CAR T cells can become exhausted by chronic antigen stimulation, characterised by upregulation of inhibitory receptors and loss of effector function[8,46]. Studies have confirmed that changes in epigenetic programming are associated with CD8 T-cell effects and memory differentiation transcription[47–49]. Inhibiting DNA methylation programmes from scratch can reduce T-cell depletion and enhance effector function[11]. DAC is a DNMT inhibitor currently approved by the FDA for use in inhibiting DNA methylation procedures. Here, we exhibit the enhanced biological function of CAR T cells reprogrammed by DAC to overcome the shortcomings of conventional CAR T cells and enhance the potential of CAR-based immunotherapy.

We optimised the dCAR T-cell production protocol based on the dosage and viability of DAC. Stable cell proliferation and viability determined the production scheme of adding 10–100 nM of DAC in synchronism with CAR gene infection to avoid the possible toxicity of DAC. Finally, a dose of 10 nM was administered in this study. Although there was no significant difference in overall proliferation, dCAR T cells showed a significant increase in CAR T cells relative to the ratio of CD4 to CD8. The reason for the increase in the CD4:CD8 T-cell ratio of the dCAR T cells may be attributed to the fact that CD4-positive cells proliferate faster after treated with DAC (Fig. 4d). Although the presence of CD4-positive cells helps improve CAR T-cell function[50], due to the enhanced proliferation of CD4 cells by DAC treatment, it may counterintuitively abrogate CD8 proliferation ex vivo. We further compared the proliferation of CD4 and CD8 cells after antigen stimulation. Although the proliferation of the CD8-positive dCAR T cells was lower than that of the CAR T cells ex vivo, this relatively lower proliferation rate was changed after antigen

stimulation (Fig. 4g). Interestingly, the expression of CD62L was more concentrated on CD4-positive CAR T cells than on CD8-positive CAR T cells after DAC treatment (Fig. 4c). In the "resting state", dCAR T cells exhibited a low cell differentiation status relative to CAR T cells from the activation-related cell phenotype to the transcription profile. Interestingly, the genes for IL-2 secretion and cell proliferation genes were upregulated. The transcription of IL-2 in nonproliferating lymphocytes is related to the demethylation of the activated IL-2 gene promoter. Therefore, dCAR T cells have enhanced secretion of IL-2 due to the promotion of IL-2 transcription by DAC and the demethylation programme of activated genes.

Conditional deletion of the de novo methyltransferase DNMT3a at an early stage of effector differentiation resulted in reduced methylation and faster re-expression of naive-associated genes, thereby accelerating the development of memory cells[11]. The downregulation of DNMT3a expression was evident in our study, suggesting that DAC has a related demethylation effect on CAR T cells. Long-term maintenance of transcription factor accessibility to gene regulatory elements is controlled in part by covalent modifications to histones and DNA that affect chromatin structure, resulting in an "epigenetic memory" of gene expression programmes in a dividing population of cells. In our study, CAR T cells from 6 donors, DNMT3a protein expression continued to be downregulated (Fig. 1f). After DAC treatment, dCAR T cells upregulated related memory transcription factors, accompanied by a significant increase in central memory T-cell expression in the "resting state".

After cytology studies, we measured the efficacy of the dCAR T cells in ALL and NHL mouse models. In "stress tests", the dCAR T cells showed strong tumour control capabilities compared to the CAR T cells. Other studies have confirmed that depleted T cells are directly related to ineffective tumour control. Consistent with the expression profile results, the dCAR T cells had a significantly lower expression of PD1/TIM3/LAG3 than the CAR T cells, and normal CAR T cells exhibited a high exhaustion phenotype with reduced expansion in the ALL model. The cell expansion observed in the in vitro experiment was also evident in the vivo experiments. The dCAR T-cell expansion capacity was far better than that of CAR T cells in the three therapeutic cell dose models, and this high expansion was maintained for more than 1 month. The disruption of DNMT3a in mature CD8 T cells results in increased memory cell differentiation and reduced terminal effector differentiation[51]. We also observed that dCAR T cells have a consistently higher memory phenotype in vivo and in vitro than CAR T cells. Especially in the tumour rechallenge experiment, the dCAR T cells rapidly expanded and controlled tumour growth after tumour cell reinoculation, and with high expression of memory phenotypes.

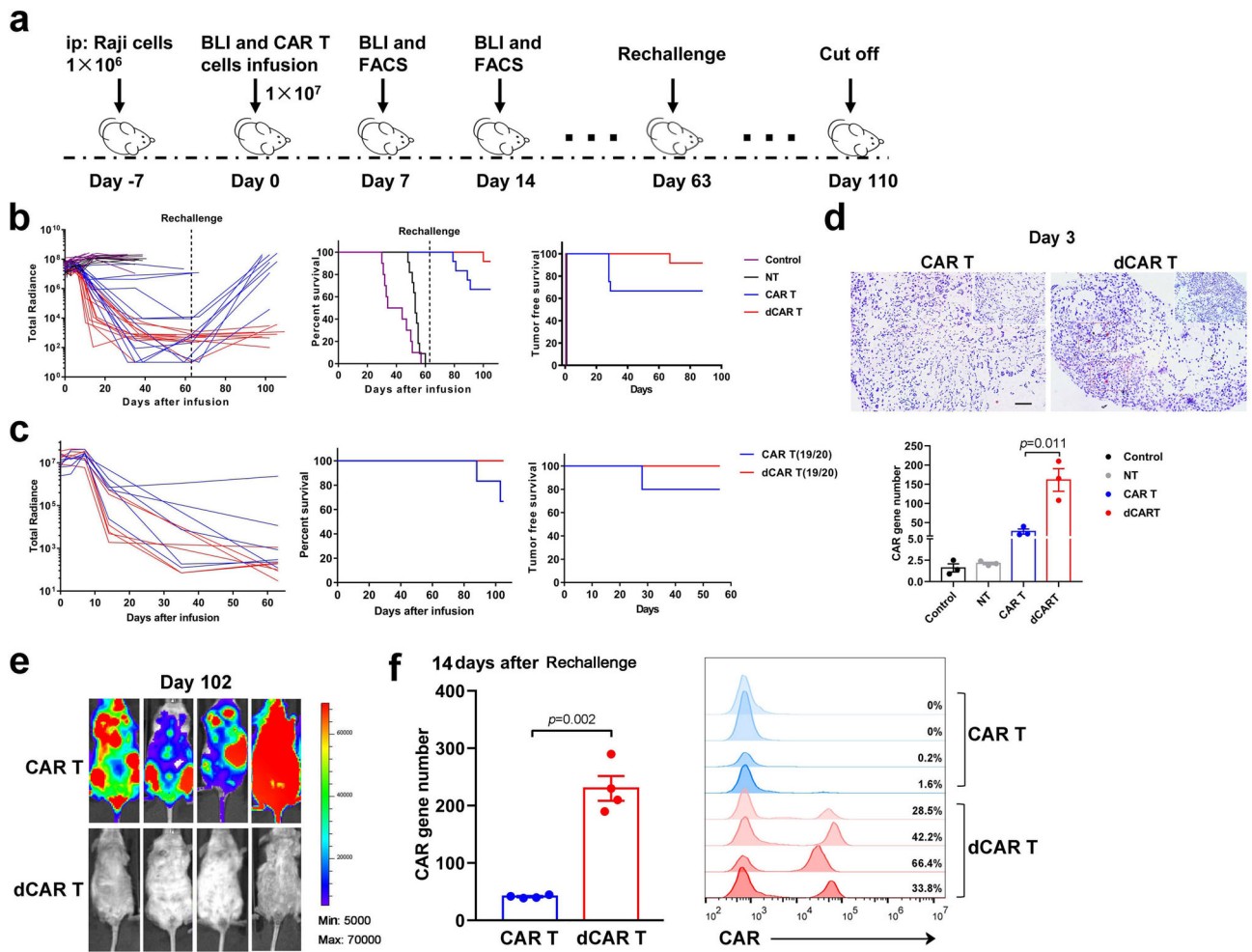

**Fig. 6 dCAR T cells exhibit enhanced tumour-free survival and effective recall responses. a** In vivo experimental layout. A total of $1 \times 10^6$ luciferase-expressing Raji cells mixed with Matrigel were intraperitoneally (i.p.) transferred to NPG mice. Tumour growth was monitored by bioluminescence (BLI). After engraftment was confirmed by increased photon activity after 7 days, $10^7$ effector cells (or the same volume normal saline as a control) were injected (i.v.), after which BLI showing the tumour burden was monitored with the schedule shown. **b, c** Left panel: Tumour burden in mice after treatment with CD19 CAR T (**b**) or CD19/CD20 tandem CAR T cells (**c**) was monitored using BLI imaging (total radiance, photons s-1 cm-2 sr-1). Survival (middle panel) and tumour-free time (right panel) were analysed by use Kaplan–Meier analysis. $P$ values were determined by the log-rank Mantel–Cox test, two-tailed. The graph is a compilation of two independent experiments. Total mouse numbers were used in (**b**) were $n = 10$ (control); $n = 10$ (NT), $n = 12$ (CAR T) and $n = 12$ (dCAR T), and in (**c**) were $n = 6$ (CAR T) and $n = 6$ (dCAR T). **d** Upper panel: Mice were sacrificed on day 3 after treatment. Paraffin tumour sections were stained with the CAR scFv-specific probe for RNA ISH. Scale bar: 100 μm. Lower panel: The CAR gene number in peripheral blood collected from each mouse at 3 days after cell infusion was measured by Q-PCR using primers specific for the transgene ($n = 3$ mice examined over two independent experiments). **e** Tumour rechallenge test: Raji cells ($2 \times 10^5$) were i.v. transferred on day 63 to the mice that achieved tumour remission after the first cell infusion. The second remission was achieved in four mice that received dCAR T-cell treatment but in no mice that received CAR T-cell treatment. **f** FACS histograms (Right panel) showing the percentage of CAR T cells and bar graph (left panel) showing the CAR gene number measured from the peripheral blood collected from four mice at 14 days after tumour cells rechallenge in two independent experiments. Data for **d** and **f** are presented as the mean ± s.e.m. $P$ values for **d** and **f** were calculated by two-tailed unpaired $t$ tests.

Recent studies have indicated that the development of memory characteristics is associated with increased antitumour efficacy and persistence in adoptively transferred T-cell subsets[34,35]. In addition, the interaction of tumour cells with CAR T cells can induce the expression of immunosuppressive molecules, such as *CTLA-4, EOMSE* and *PD1*, through signalling pathways that lead to T-cell "exhaustion" or "dysfunction", and the exhaustion of CAR T cells under long-term antigen stimulation is also a factor in disease recurrence or resistance[7]. We provide direct evidence that dCAR T cells can maintain a strong tumour lytic capacity for target cells even at very low target ratios. The dCAR T cells upregulated cell proliferation, activated genes, and downregulated inhibitory factors after stimulation by tumour cells in vitro. Compared with CAR T cells, dCAR T cells maintained a relatively

low cell differentiation state (Fig. 5). We further analysed the phenotype and gene transcription profile of tumour-infiltrating CAR T cells, and the dCAR T cells maintained a higher expansion and activation capacity in the tumour on the days 7 and 14 (Figs. 9 and 10). The differentially expressed genes of the dCAR T cells compared to the CAR T cells are enriched in the functions of proliferation, cytokine secretion, cytotoxicity and memory. Importantly, the dCAR T cells are in any state, including "resting state", antigen stimulation in vitro state or tumour infiltration state. The memory-related phenotypes of the dCAR T cells were always significantly higher than those of the CAR T cells and consistent with the phenotypic results, memory transcription factors including *TCF7, BCL6, LEF1* and the important memory-related gene *IL7R* were highly expressed in the dCAR T cells. The

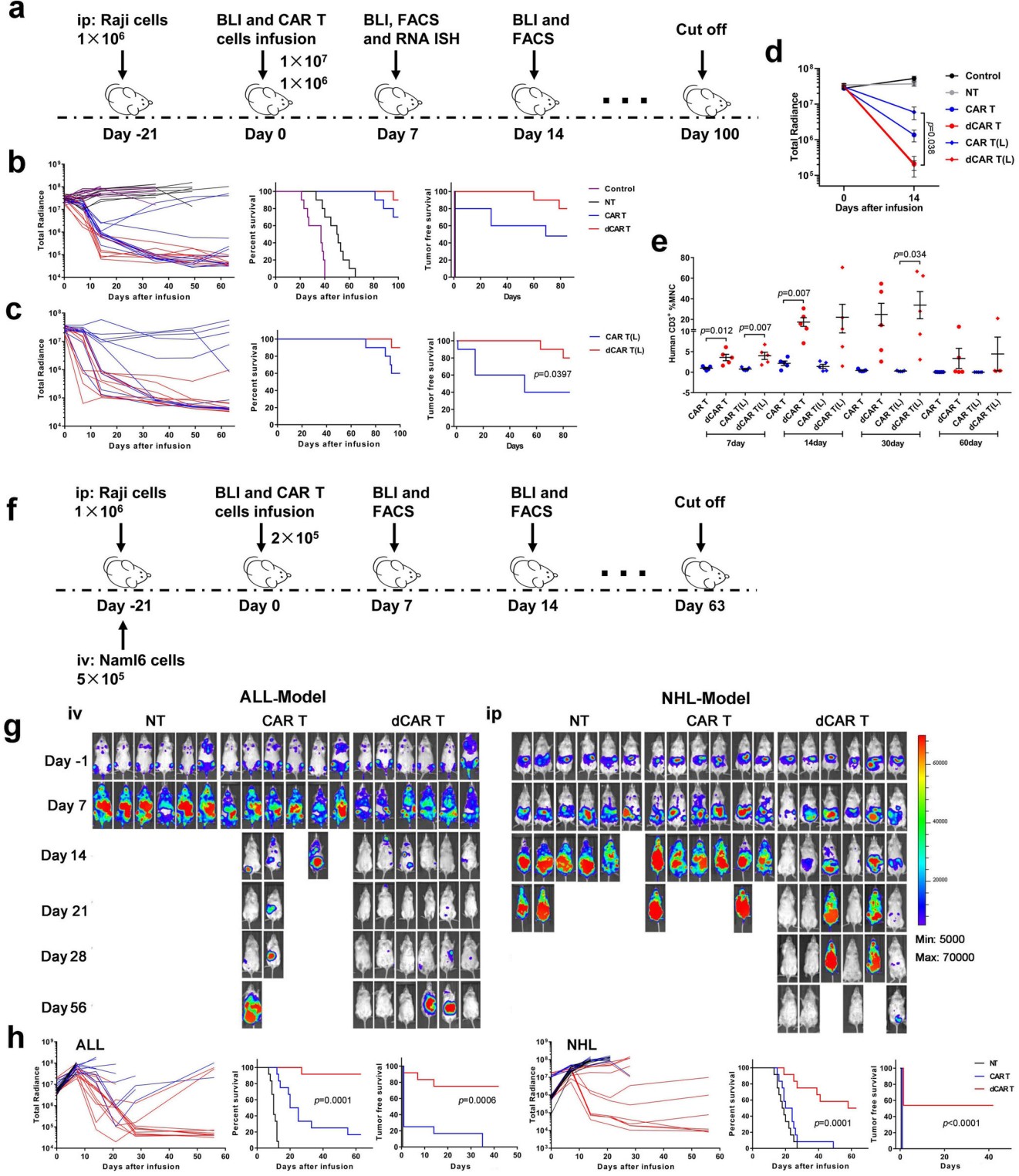

phenotype of the dCAR T cells seems to be more inclined to activate memory or memory precursor cells[52], which can maintain better expansion capacity and reduce exhaustion and apoptosis.

In summary, we demonstrated that CAR T cells underwent DNA reprogramming after DAC treatment, which induced significant sustained cell expansion, cytotoxicity and cytokine production and reduced exhaustion after antigen exposure (Fig. 10f). Especially shown by the "stress test", the dCAR T cells at very low

doses could efficiently control tumours with a very large tumour burden. Importantly, the dCAR T cells maintained a higher proportion of cells with a memory phenotype than did the CAR T cells under long-term tumour stimulation. The genetic modification of CAR T cells to enhance their memory phenotype[18,20,53] or reduce T-cell exhaustion[54] is widely studied, with the aim of transforming CAR T cells to enhance their antitumour effects or reduce tumour recurrence. The addition of demethylating drugs may become a convenient and economical

**Fig. 7 dCAR T cells display enhanced in vivo antitumour activity against a large tumour burden at the lowest dose examined. a** In vivo experimental layout. A total of $1 \times 10^6$ luciferase-expressing Raji cells mixed with Matrigel were intraperitoneally (i.p.) transferred to NPG mice. Tumour growth was monitored by BLI. After 21 days, effector cells (or the same volume normal saline as a control) were injected (i.v.), after which the tumour burden was assessed by monitoring BLI with the schedule shown. **b, c,** Left panel: Tumour burden in mice treated with $1 \times 10^7$ CD19 CAR T or dCAR T cells (**b**) and $1 \times 10^6$ CD19 CAR T or dCAR T cells (**c**) was monitored using BLI imaging. Survival (middle panel) and tumour-free survival (right panel) were analysed by Kaplan–Meier analysis (ten mice examined over two independent experiments). "L" was defined as $1 \times 10^6$ cell treatment group. **d** Tumour burden of the mice ($n = 5$) were assessed by determining BLI radiance on days 0 and 14. **e** The percentage of CAR T cells in the PB was used to evaluate the expansion and persistence of CAR T cells at the indicated time points ($n = 5$). MNC mononuclear cell. Data for **d** and **e** are presented as the mean ± s.e.m. $P$ values for **d** and **e** were calculated by two-tailed unpaired $t$ tests. **f** In vivo experimental layout. NHL model: A total of $1 \times 10^6$ luciferase-expressing Raji cells mixed with Matrigel were i.p. transferred to NPG mice. ALL model: A total of $5 \times 10^5$ luciferase-expressing Nalm-6 cells were i.v. transferred to NPG mice. After 21 days, $2 \times 10^5$ effector cells were injected (i.v.) (12 mice examined over two independent experiments). **g** Representative graph: BLI images showing mouse tumour burdens at the indicated time points of one experiment. **h** (left panel: ALL) and (right panel: NHL). Left panel: Tumour burden in mice after treatment with CD19 CAR T or dCAR T cells was monitored using BLI imaging. Survival (middle panel) and tumour-free survival (right panel) were analysed by Kaplan–Meier analysis. $P$ values for **b**, **c** and **h** were determined by the log-rank Mantel–Cox test, two-tailed.

means to consistently achieve this goal. Preparations for clinical studies that utilise DAC reprogramming of CAR T cells are underway.

## Methods

**Manufacturing of CAR T and dCAR T Cells.** CAR or tandem CAR was generated by linking the CD19 scFv derived from the FMC63 monoclonal antibody (mAb)[55] or CD20 scFv derived from the Leu-16 mAb[56] in frame with the hinge and transmembrane domains of CD8, and the cytoplasmic domains of 4-1BB and CD3ζ. The CAR constructs were verified by DNA sequencing and synthesised and cloned into the pRRLSIN lentiviral plasmid backbone under the regulation of a human EF-1α promoter. Then, 293T cells were transiently transfected with the pRRLSIN vector and the psPAX2 and pMD2G packaging plasmids. The lentiviral supernatants were collected and stored at −80 °C.

CAR T cells were generated by infection of activated T cells with CAR-encoding lentivirus. In detail, peripheral blood mononuclear cells (PBMCs) were collected from patients by leukapheresis. Primary T cells derived from the PBMCs were activated using Dynabeads Human T-Activator CD3/CD28 magnetic beads (Invitrogen) at a bead:cell ratio of 1:1 and cultured in X-VIVO 15 medium (Lonza) supplemented with 300 U/ml recombinant human IL-2 (PeproTech). After 2 days, RetroNectin-coated plates (TaKaRa) were loaded with viral supernatants by centrifuging the plates at $2000 \times g$ for 2 h at 4 °C. Activated T cells were added to the plates and centrifuged at $800 \times g$ for 10 min, and polybrene (Sigma) was added to increase the infection efficiency. 5-aza-2'-deoxycytidine (DAC, Sigma) was added to the culture of dCAR T cells. The following day, the transduced cells were collected and transferred to new flasks for further expansion. After transduction, the cells were expanded ex vivo in the presence of IL-2 added three times weekly until the specified cell dose was achieved. Purified CAR T cells were obtained by sorting (magnetic beads) at 3 days post-lentivirus infection. The Dynabeads were magnetically removed at the end.

**Cell line generation and maintenance.** The Burkitt lymphoma cell line Raji, the B lymphocyte leukaemia cell line Nalm-6 and the chronic myelogenous leukaemia cell line K562 were purchased from ATCC (Manassas, VA). These cells were then infected with the pLenti-CMV-luc2-IRES-Puro virus to express firefly luciferase. Cells stably expressing firefly luciferase were established by puromycin selection. All cell lines were cultured in RPMI-1640 medium (Gibco) supplemented with 10% foetal bovine serum (HyClone), 2 mM L-glutamine (Gibco) and 100 U/mL penicillin/streptomycin (HyClone).

**Flow cytometry analysis.** For CAR staining, CAR T cells or peripheral blood samples were incubated with a biotin-SP-AffiniPure F(ab')2 fragment-specific goat anti-mouse IgG antibody (Jackson ImmunoResearch, 1:100 dilution), followed by incubation with streptavidin-phycoerythrin (PE, 1:500 dilution) or streptavidin-fluorescein isothiocyanate (FITC, 1:500 dilution) (BD Biosciences).

For a degranulation assay, CAR T cells were co-cultured with Raji cells at an E:T ratio of 1:1 in X-VIVO 15 medium containing an anti-human CD107a antibody (BD Biosciences) for 0.5 or 1 h, followed by incubation with a Golgi Plug protein transport inhibitor (BD Biosciences) for 3 h. In addition, the following anti-human antibodies were also used in this study:: CD3 (chlorophyll protein complex PerCP and APC, 1:50 dilution), CD4 (PE and FITC, 1:50 dilution), CD8 (PE, FITC and APC-cy7, 1:50 dilution), CD45RO (APC and FITC, 1:100 dilution), CD62L (APC, 1:100 dilution), PD1 (PE and APC, 1:100 dilution), TIM3 (FITC and PE, 1:100 dilution), LAG3 (BV510, 1:100 dilution), HLA-DR (APC, 1:100 dilution), Ki67 (PE, 1:100 dilution), CD25 (APC, 1:100 dilution), EOMES (FITC, 1:100 dilution), Foxp3 (PE, 1:100 dilution), IL-17A (PE, 1:100 dilution) were purchased from BD Biosciences. Isotype-matched control mAbs were applied in all the procedures.

FACS data were analysed by Cytexpert software (version 1.1.10.0, BECKMAN COULTER) and FlowJo software (Version 10.0.7, FlowJo, Ashland, OR).

**In vitro functional assays.** To test the lysis of target cells, CAR T cells were co-cultured with target cells at different E:T ratios in 96-well plate. One hundred microlitres of 2×D-Luciferin solution (300 µg/ml) were added to each well, and the signals were measured after 2–5 min by Varioskan™ LUX (Thermo Fisher). The lysis was calculated by the following formula: 1-([value of sample]-[value of negative control])/([value of positive control]-[value of negative control]). In addition, cytokine levels in the culture supernatants were quantified by enzyme-linked immunosorbent assay (ELISA; Novus Biologicals) and Luminex assay.

The continuous cytotoxicity assays of CAR T cells were monitored using an xCELLigence Real-Time Cell Analyzer-Multiple Plate system (Agilent Technologies). This platform measures surviving adhesive target cells in real time. First, 10,000 Raji cells, CD40 antibody and 100 µl of culture medium were added to each well of a 96-well plate (E-Plate 96, Agilent Technologies, USA), which was then placed into the Real-Time Cell Analyzer-Multiple plate instrument at 37 °C with 5% $CO_2$. After 24 h, 2000 effector T cells suspended in 100 µl of culture media was replaced. After 72 h, 100 µl of fresh culture medium was replaced. Cell index values were calculated using the RTCA Software. The cell index represented changes in electrical impedance and reflected the number of surviving target cells on biocompatible microelectrode surfaces. For real-time monitoring, the cell index was read automatically every 15 min. Cell index data in each group represented the mean value from three wells.

To obtain dynamic images of lysis of target cells, CAR T cells were co-cultured with target cells at an E:T ratio of 1:30 in 96-well plate. CAR T cells were labelled with NucLight Green live-cell labelling reagent (Essen BioScience). IncuCyte® Annexin V Red Reagent (Essen BioScience): solubilise Annexin V by adding 100 µl of complete medium or PBS. The reagents may then be diluted in complete medium containing at least 1 mM $CaCl_2$ for a dilution of 1:50 (4×) final assay concentration, 50 µl per well. Use the IncuCyte® live-cell analysis system to automated detect and select quantitation of tumour cell death in real time.

In a long-term proliferation experiment, CAR T cells were treated with different doses of DAC at a density of $10^6$ cells per ml in 24-well plates. The total cells and viability rate were counted at scheduled times, and CAR expression was determined weekly by flow cytometry.

**Detection of DNMT3a by western blot analysis.** The purified CAR T and dCAR T cells were obtained by magnetic bead sorting, purity was determined by flow cytometry. The purified CAR T and dCAR T cells were lysed in RIPA buffer (Cell Signaling Technology) with protease inhibitor. The protein concentration was quantified by a Bio-Rad protein assay. The total protein was separated on a 4–12% SDS-PAGE gel followed by standard immunoblotting with antibody to β-actin (Abcam, 1:1000 dilution) and antibody to DNMT3a (Abcam, 1:2000 dilution).

**Quantitative Real-time PCR.** Real-time quantitative polymerase chain reaction (Q-PCR) was used to assess the level of the CAR fusion genes. A 153-bp (base pair) fragment containing portions of the CD8α chain and adjacent 4-1BB chain was amplified using forward primer 5'-GGTCCTTCTCCTGTCACTGGTT-3' and reverse primer 5'-TCTTCTTCTTCTGAAATCGGCAG-3' to detect the CAR gene; amplification of β-actin was used as an internal control and for normalisation of DNA quantities. Q-PCR was performed using SYBR Green Master Mix (Toyobo, Japan) and run on an ABI prism 7500 sequence detection system (Applied Biosystems). A 7-point standard curve that consisted of 100–108 copies/µL plasmid DNA containing the CAR gene was prepared.

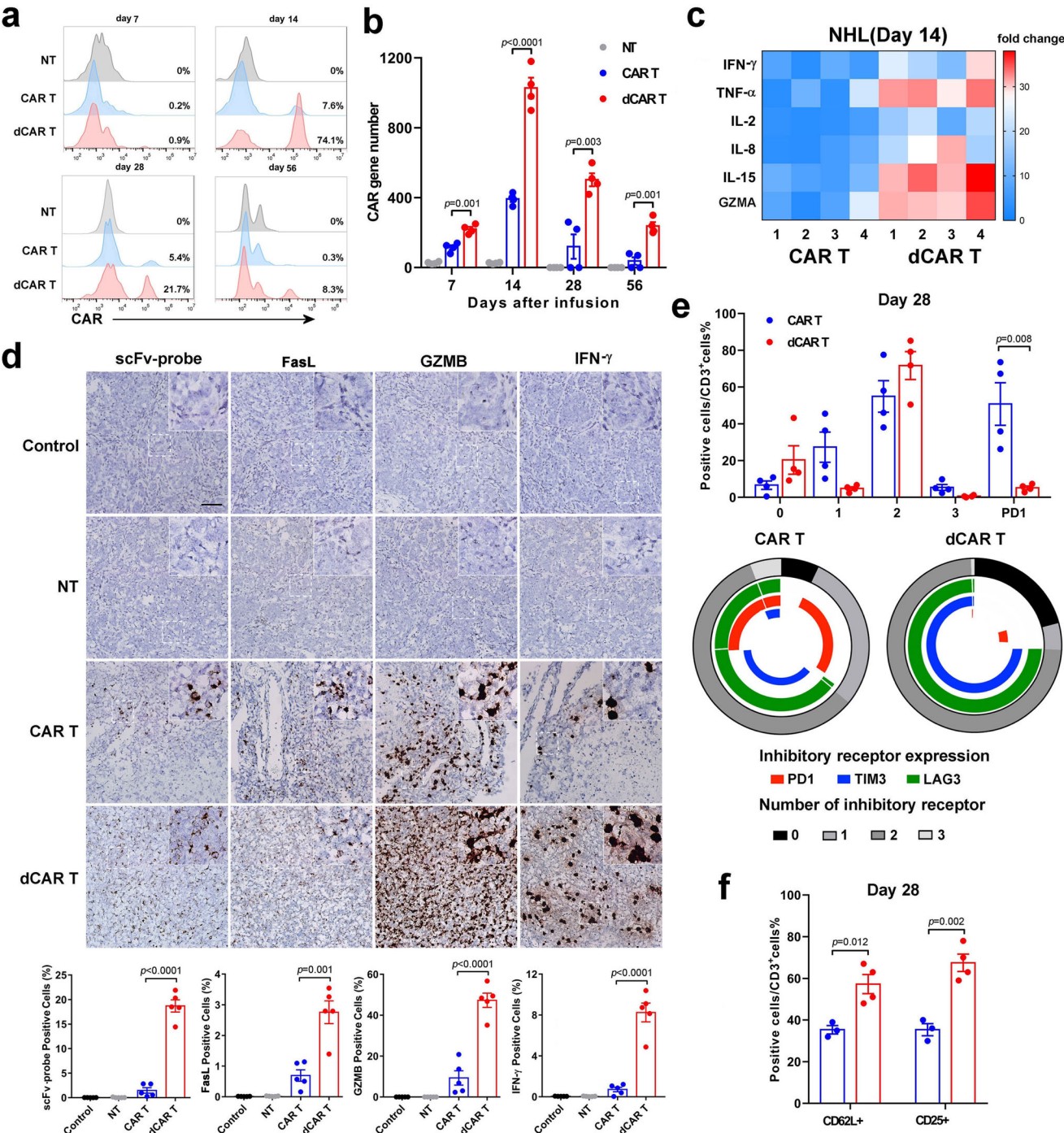

**Fig. 8 Enhanced in vivo antitumour function and reduced exhaustion of dCAR T cells. a** Representative histograms showing the percentage of CAR T cells in the PB at the indicated time points are presented and were used to evaluate the in vivo expansion of CAR T cells. **b** The CAR gene number from PB samples taken from mice-bearing tumours from Raji cells treated with $1 \times 10^6$ CAR T cells in each group collected at serial time points after cell infusion was measured by Q-PCR using primers specific for the transgene ($n = 4$ mice over two independent experiments). **c** Cytokines from the sera of venous blood samples from mice from ALL model experiments ($n = 4$ mice over two independent experiments) collected at 14 days after cell infusion were measured by FACS. All data are the fold change in the value measured in the test group compared to that in the control group. Colours indicate fold changes of cytokine levels. **d** Upper panel: Mice-bearing tumours from Raji cells treated with $1 \times 10^6$ CAR T cells were sacrificed on day 14 after treatment. Paraffin tumour sections were stained with CAR scFv-, granzyme B- (GZMB-), IFN-γ- and FasL-specific probes for RNA ISH. Scale bar: 100 μm. Lower panel: The positive cell count value is the mean value from ten randomly selected fields in each slice (five slices from three mice in two independent experiments). **e** Upper panel: Statistical analysis of differences in the ratio of cells expressing the non, 1, 2 or 3 phenotypes (PD1, LAG3 and TIM3) of the CAR T cells. Lower panel: Expression of PD1, LAG3 and TIM3 in CAR T cells. **f** Expression of CD3/CD62L and CD3/CD25 on CAR T cells. To obtain the data in (**e**) and (**f**), cells were collected from the bone marrow of Raji tumour cell-bearing mice treated with $1 \times 10^6$ CAR T cells on day 28 after cell infusion ($n = 4$ over two independent experiments). Data for **b**, **d**, **e** and **f** are presented as the mean ± s.e.m. *P* values for **b**, **d**, **e** and **f** were calculated by two-tailed unpaired *t* test.

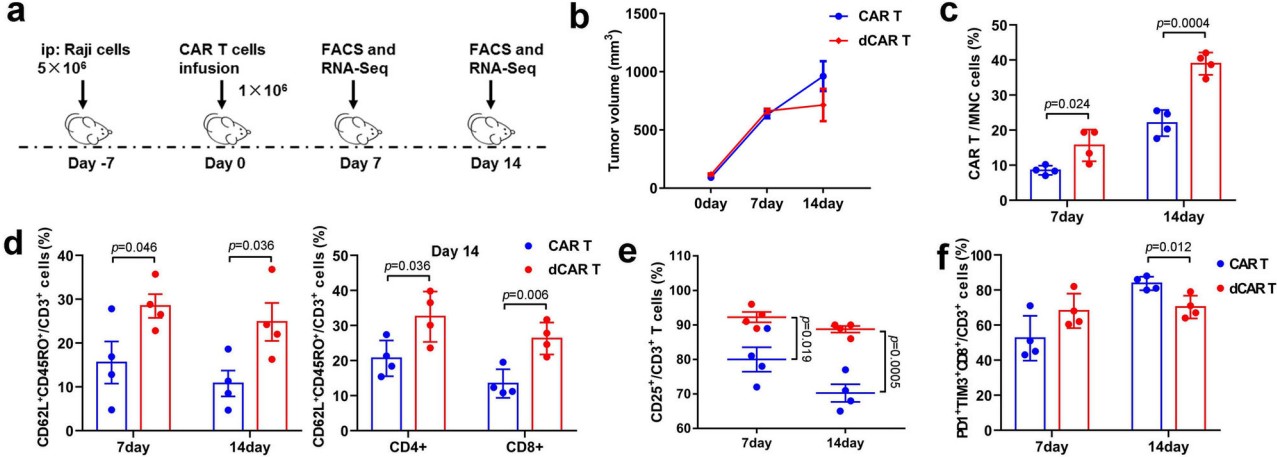

**Fig. 9 Tumour-infiltrating dCAR T cells exhibit enhanced proliferation, activation and improved memory-phenotype under chronic tumour antigen-stimulation in vivo. a** In vivo experimental layout. A total of $5 \times 10^6$ luciferase-expressing Raji cells mixed with Matrigel were intraperitoneally (i.p.) transferred to NPG mice. Tumour growth was monitored by bioluminescence (BLI). After 7 days, $1 \times 10^6$ dCAR T or CAR T cells were injected (i.v.) following the schedule shown. Tumours were obtained from five mice on sacrificed days 7 and 14 after cell infusion to measure tumour volume and isolate tumour-infiltrating dCAR or CAR T cells. **b** Cumulative data are shown the tumour burden of the mice, as assessed by tumour volume on days 0 and 14 ($n = 5$ mice examined over two independent experiments). Data are presented as the mean ± s.e.m. **c** The percentage of CAR T cells among the total number of MNC of the tumour ($n = 4$). **d** Expression of CD62L/CD45RO in tumour-infiltrating CD3-positive, CD4-positive and CD8-positive CAR T cells on days 7 and 14 after cell treatment ($n = 4$). **e** Expression of CD25 in tumour-infiltrating CAR T cells ($n = 4$). **f** Expression of PD1/TIM3/CD8 among tumour-infiltrating CAR T cells on days 7 and 14 after cell treatment ($n = 4$). The data in **c–f** are presented as the mean ± s.e.m. P values for **c–f** were calculated using a two-tailed unpaired t test.

**Measurement of serum cytokine levels**. Cytokine from the mice serum and the cultural supernatant were evaluated by Luminex assay according to the manufacturer's instructions.

**Mouse studies**. For the in vivo study of CAR T-cell function, we established intraperitoneally injected Raji-luc lymphoma and intravenously injected Nalm-6luc ALL models, in which female NPG mice (NOD-Prkdc$^{scid}$Il2r$^{gnull}$/Vst, stock No. VS-AM-001, Beijing Vitalstar Biotechnology Co.,Ltd.) aged 4–6 weeks were used. The mice received an injection of 100 μl each time and all the cells would be adjusted to an appropriate volume prior to administration. CAR T, dCAR T and NT cells were resuspended in PBS and injected into mice through the tail vein. Tumour burdens were monitored and quantified in vivo by BLI on NightOwl II(LB 983, Berthold) Platform, BLI data were analysed using indiGO software (Berthold) and the BLI signal was reported as the average flux (photons per second/area [mm$^2$]). All animals were anaesthetised with isoflurane gas.

For tracing the CAR T cells and released cytokines in peripheral blood, ~200 μl of blood was taken through the canthus for subsequent flow analysis. To analyse the copy numbers and function of intratumoral CAR T cells, the tumour masses were obtained by excision and fixed in formalin for subsequent RNA ISH experiments, and bone marrow was obtained by excision for Q-PCR and flow cytometry. Animals were housed in the Chinese People's Liberation Army General Hospital Laboratory Animal Center facilities where temperature, humidity and illumination were maintained according to the Institutional Animal Care and Use Committee of Chinese People's Liberation Army General Hospital guidelines. All animal studies were performed in compliance with institutional regulations for animal use. Mouse handlers were blinded to group allocation.

**Isolation of tumour-infiltrating CAR T cells for subsequent analyses**. On days 7 and 14 after cell infusion, five to eight mice were killed, and tumours were removed. Tumours were collected, pooled together by the group, homogenised, and then dissociated using the MACS Miltenyi Mouse Tumour Dissociation Kit (Miltenyi Biotec) according to the manufacturer's instructions. Purified CAR T cells were obtained by sorting (magnetic beads, Miltenyi Biotec) according to the manufacturer's instructions. For flow cytometry analyses, 500,000 cells for each group were sorted from the isolated tumour-infiltrating CAR T cells. For RNA-seq, two technical replicates of $1 \times 10^6$ cells each were sorted from the isolated tumour-infiltrating CAR T cells.

**Methylation sequencing data analysis**. DNA was isolated from T cells in all studies using DNeasy Kit (Qiagen). The purity and concentration of DNA were estimated using Nanodrop 2000 (ThermoScietific). Approximately, 500 ng of genomic DNA from each sample was used for sodium bisulfite conversion using the EZ DNA methylation Gold Kit (Zymo Research, USA) following the

manufacturer's standard protocol. Genome-wide DNA methylation was assessed using the Illumina Infinium HumanMethylation850K BeadChip (Illumina Inc, USA) according to the manufacturer's instructions. The array data (.IDAT files) was analysed using package in R software v3.5. for deriving the methylation level. The methylation status of all the probes was denoted as beta (β) value, which is the ratio of the methylated probe intensity to the overall probe intensity (sum of methylated and unmethylated probe intensities plus constant α, where α = 100). Generalised linear models (v3.36.2) were used to identify significantly differentially CpG sites. A P value of 0.05 or less was regarded as a significant difference.

**RNA-sequencing data processing**. Raw data of fastq format were first processed, and clean data (clean reads) were obtained by removing reads containing adapters, reads containing poly-N and low-quality reads from raw data. The quality control of raw fastq data was by FastQC. The index of the reference genome was built using Hisat2 v2.0.5 and paired-end clean reads were aligned to the reference genome using Hisat2 v2.0.5. Pair-end reads were trimmed using TrimGalore (https://github.com/FelixKrueger/TrimGalore). Cleaned reads were mapped to UCSC Human GRCh38/hg38 (http://genome.ucsc.edu/). The mapped reads of each sample were assembled by StringTie (v1.3.3b). Trimmed reads with a mapping quality exceeding 20 were counted by HTSeq (Hisat2 v2.0.5). FPKM of each gene was calculated based on the length of the gene and read count mapped to this gene.

**RNA-sequencing analysis**. Prior to differential gene expression analysis, the read counts were adjusted for each sequenced library by the edgeR program package through one scaling normalised factor. The DEseq2 package version 1.16.1 in the statistical analysis software R/Bioconductor was used to analyse the normalised RNA-seq data. Generalised linear models with empirical Bayes approach were used to identify significantly differentially expressed genes. A P value of 0.05 or less was regarded as a significant difference. For clustering, we clustered different samples to see the correlation using the hierarchical clustering distance method with the heatmap function using the silhouette coefficient to adapt the optimal classification with default parameter in R.

**GO enrichment analysis of differentially expressed genes**. Gene Ontology (GO) enrichment analysis of differentially expressed genes was implemented by the cluster Profiler R package, in which gene length bias was corrected. GO terms with corrected P values less than 0.05 (or P values<0.05) were considered significantly enriched by differentially expressed genes.

**Gene set enrichment analysis (GSEA)**. Raw data were normalised in two replicates for GSEA analysis. The genes were ranked according to the degree of differential expression in the two samples, and then the predefined gene set was

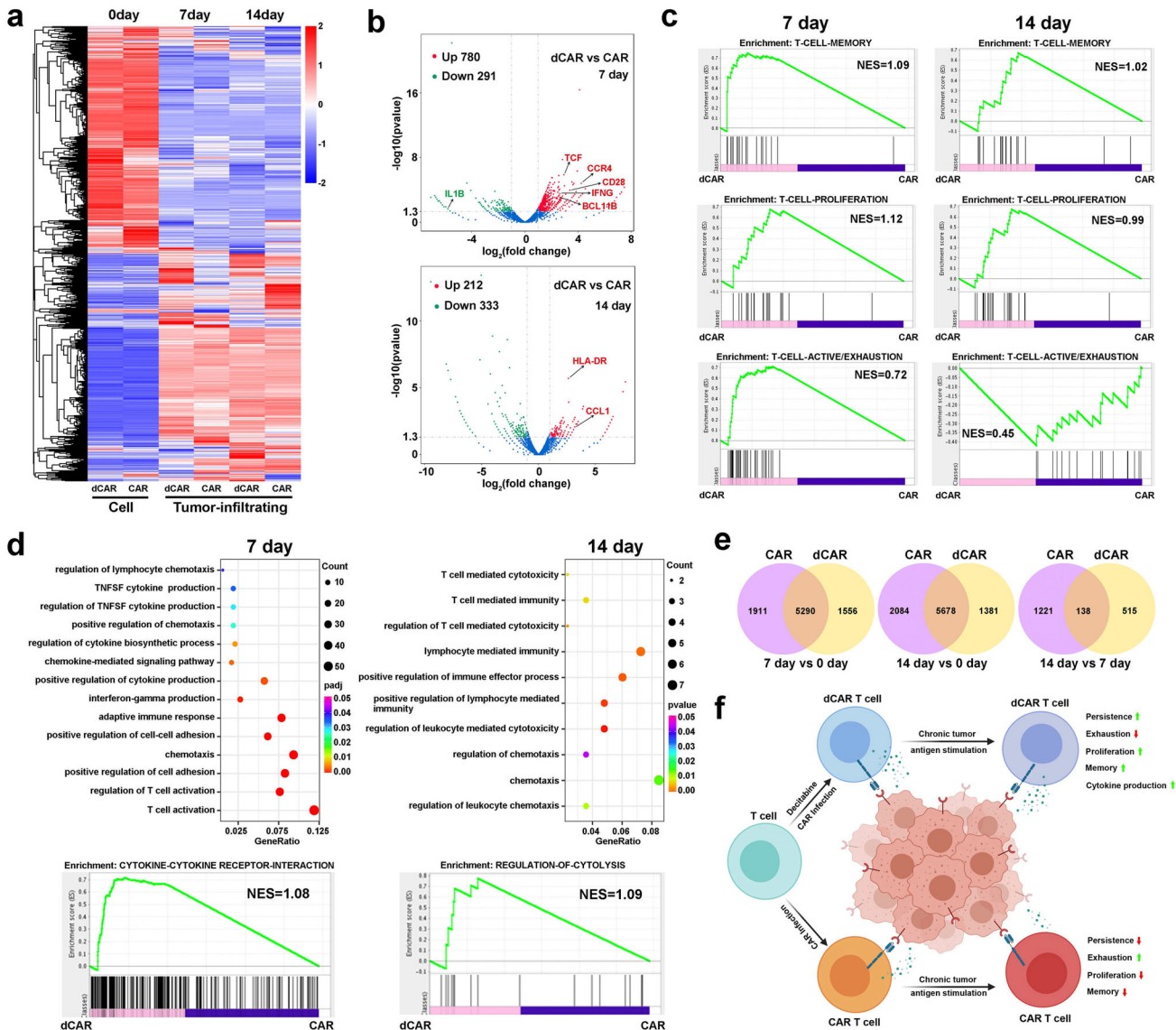

**Fig. 10 Tumour-infiltrating dCAR T cells show increased effector and memory function compared to tumour-infiltrating CAR T cells. a** Hierarchical clustering of the RNA-seq analysis results shows significantly differentially expressed genes between tumour-infiltrating dCAR T cells versus tumour-infiltrating CAR T cells at different time points after cell infusion. Cluster columns are displaying in groups ($n = 2$ replicates per group). **b** The volcano plot illustrates fold changes in gene expression in tumour-infiltrating dCAR T cells compared with tumour-infiltrating CAR T cells. Significantly upregulated genes are shown in red, significantly downregulated genes are shown in blue, and genes whose difference in expression was statistically nonsignificant are shown in grey. Values are presented as the log2 of the tag counts. **c** Representative GSEA enrichment plot, demonstrating significantly upregulation of memory-related genes and proliferation-relative genes in tumour-infiltrating dCAR T cells versus tumour-infiltrating CAR T cells on day 7 and day 14, significantly upregulation of active/exhaustion-relative genes on day and downregulation of active/exhaustion-relative genes on day 14 in tumour-infiltrating dCAR T cells versus tumour-infiltrating CAR T cells. **d** GO functional clustering of genes that were significantly upregulated for biological processes in tumour-infiltrating dCAR T cells versus tumour-infiltrating CAR T cells and representative GSEA enrichment plot. **e** Venn diagram representing the overlap of the different genes ($P < 0.05$) in the CAR T and dCAR T cells before and after cell infusion in vivo. **f** The potential functional advantages of dCAR T cells compared to CAR T cells in the antitumour process. Significantly differentially expressed genes in (**a**) to (**e**) were calculated the Wald test (as implemented in DESeq2) ($P$ value <0.01, fold change (log2 scale) $\geq 1$ or $\leq -1$). Image **f** created with BioRender.com.

tested to determine if they were enriched at the top or bottom of the list. We use the local version of the GSEA analysis tool http://www.broadinstitute.org/gsea/index.jsp, and the GO and KEGG sets were used for GSEA independently.

**RNA in situ hybridisation (RNA ISH).** Paraffin-embedded tissue sections cut at a 5-µm thickness were processed for RNA in situ detection using the RNAscope® 2.5 HD Reagent Kit-BROWN (ADC, Advanced Cell Diagnostics) according to the manufacturer's instructions. Briefly, the tissue sections were baked for 30 min at 60 °C, deparaffinized with xylene, and dehydrated with ethanol. The tissue was pre-treated with RNAscope hydrogen peroxide for 15 min at room temperature, RNAscope Target Retrieval Reagent for 8 min at 98 °C, and RNAscope Protease

Plus for 15 min at 40 °C. The RNAscope Target Probes used included a customised scFv-, FasL-, IFNγ- and a Granzyme B-specific probe (ADC, Advanced Cell Diagnostics; Probe sequence list are shown in Supplementary Table 1). Positive and negative control probes were used for each experiment according to the manufacturer's instructions. The probes were added to the tissue and hybridised for 2 h at 40 °C. A series of amplification steps were performed using the instructions and reagents provided in the RNAscope 2.5 HD Detection Reagent Kit-BROWN. The tissue was counterstained with Gill's haematoxylin for 25 s at room temperature, followed by mounting with neutral resin. Images were acquired with an Olympus microscope. For quantification of compartments by ISH, at least 1000 cells were counted in representative areas of the tumours.

**Bisulfite modification, methylation-specific PCR (MSP) and bisulfite sequencing**. DNA from CAR T and dCAR T cells cultured for 10 days were sequenced by sodium bisulfite treatment in this study. Bisulfite-treated DNA was amplified using primers flanking the targeted regions, including the MSP amplified region and the transcriptional start site. *PD1* gene sequencing primers were as follows: 5′-TTGGGYYGGTGTTATAATTGGG-3′ (forward) and 5′-CTAAACCTA CCACAACACCC-3′ (reverse). *TIM3* gene sequencing primers were as follows: 5′-TGTGATTGTAGATTTGGTAGTG-3′ (forward) and 5′-CCTCTATACAACAC CATTATATC-3′ (reverse). PCR cycle conditions were as follows: 95 °C × 5 min for 1 cycle; 35 cycles ×; (95 °C × 30 s, 55 °C × 30 s, 72 °C × 40 s); 72 °C × 5 min for 1 cycle. PCR products were gel purified and cloned into pCR2.1 vectors according to the manufacturer's protocol (Invitrogen). Colonies were grown on agar plates and randomly selected. Plasmids were then isolated and purified using Wizard miniprep kits (Promega, Shanghai, China). Integrated PCR fragments were confirmed with EcoRI digestion (New England Biolabs, Beverly, MA, USA) and the cloned PCR fragments were sequenced with the M13 reverse primer via automated sequencing (BGI Sequencing, Beijing, China).

**Statistical analysis**. All statistical analyses were performed using the Prism 7 (GraphPad) software. Comparisons of two groups were determined by two-tailed parametric t tests for unpaired or paired data. A log-rank Mantel–Cox test was used to compare survival differences between the groups. P values <0.05 were considered to be statistically significant. The statistical test used for each figure is described in the corresponding figure legend.

**Study approval**. Human materials in this study were approved by the Institutional Review Board at the Chinese People's Liberation Army General Hospital. Human immune cells were isolated from PBMCs from healthy donors who provided written informed consent before enrollment in the study. All animal experiments were approved by the Institutional Animal Care and Use Committee of Chinese People's Liberation Army General Hospital. All the procedures of human materials and animal experiments were performed in accordance with the guidelines of the Institutional Review Board and Animal Care and Use Committee of Chinese People's Liberation Army General Hospital.

**Reporting summary**. Further information on research design is available in the Nature Research Reporting Summary linked to this article.

## Data availability

The RNA-sequencing data generated in this study have been deposited in the GEO (Gene Expression Omnibus) database under accession code GSE156207. The DNA Methylation data generated in this study have been deposited in the GEO database under accession code GSE161506. The remaining data are available within the Article, Supplementary Information or available from the authors upon request. Source data are provided with this paper.

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

## Acknowledgements
This work was supported in part by the National Natural Science Foundation of China (31991171, and 81830002), the Science and Technology Planning Project of Beijing City (No. Z151100003915076) and the National Key Research and Development Program of China (No. 2017YFC0909803 to Y.W., and No. 2019YFC1316205 to J.N.). We thank Professor Xin Lin of Tsinghua University and Professor Mingzhou Guo of Chinese People's Liberation Army General Hospital for support of data analysis.

## Author contributions
Y.W. designed experiments, analysed the data, created the figures and wrote/edited the paper. C.T., H.D. and Z.W. contributed to edited the paper. C.T., X.H., Z.W., Y.G., D.C., J.W. and D.T. performed experiments. C.T., X.H. and Z.L. analysed the data. Q.M., X.L., L.D., J.N. and Y.Z. assisted in the experiments. W.H. supervised the project, contributed to the design of the experiments, and edited the paper. All authors critically read and approved the paper.

## Competing interests
The authors declare no competing interests.
