## [Peer Review File · Nature Communications]

Reviewers' comments:

Reviewer #1 (Remarks to the Author): expertise in CAR-T

In this manuscript, authors report that ex vivo CAR T cell expansion using low dose DAC endows CAR T cells with enhanced anti-tumor activity in vitro and in vivo using Raji and Nalm6 models, increased proliferative capacity, persistence, cytokine response, reduced expression of exhaustion-associated markers, and a phenotype resembling that of central memory T cells. Authors show methylome and RNA-seq analyses of CAR and dCAR T cells pre- and post-antigen stimulation and argue that DAC-induced reprogramming is stable over time and magnified after antigen exposure. They conclude that demethylating agents during ex vivo expansion is a convenient and economic strategy to improve CAR T cell therapy in humans.

Developing methods to improve CAR T cell function by modifying ex vivo expansion conditions is an important topic for the field, and has been the focus of other publications in the last few years (ex. small molecule AKT or PI3K inhibitors augment CAR-T responses). However, I have several major concerns regarding this manuscript. First and foremost, several major experiments that support the manuscript's overall conclusion appear to have been conducted only once. Many of these are outlined below in the specific points, but include critical in vivo experiments. These must be repeated and exhibit reproducibility. Second, several primary and extended figures lack statistical rigor. Examples include methylomic and transcriptomic analyses (ex. Figures 1i, 3a, 3d), which the authors use to support their claim of DAC-mediated CAR T reprogramming. Careful examination of heatmaps in Extended Fig. 4a-b display a concerning degree of variability. Additional replicate samples and statistics should be used to fully validate the degree to which DAC induces cell reprogramming. Third, a large portion of the beginning of the manuscript is devoted to describing differential effects of DAC on CD4s versus CD8s. In fact, low dose DAC enhanced CD4 proliferation, but counterintuitively abrogated CD8 proliferation ex vivo. The mechanism by which this occurs should be addressed. Further, methylomic and transcriptomic differences between CD4s and CD8s, as well as those potentially observed in vivo, are not addressed. This is a critical factor to understanding the effects of DAC on CAR T cell function and deserves further interrogation. Finally, authors make the argument that ex vivo DAC treatment reduces the capacity of CAR T cells to become exhausted in vivo. This claim is supported mainly by FACS data, indicating a decrease in the frequency of PD-1+ cells in dCAR T cells (but not LAG-3 or TIM-3). Why is PD-1 only affected by DAC pre-treatment? It's possible that the increased proliferation and enhanced functionality of dCAR T cells in vivo is independent of exhaustion and perhaps related to suppression enforced by the tumor microenvironment. Careful methylomic and/or transcriptomic analyses of in vivo dCAR T cells is necessary to determine whether these cells are truly resistant to T cell exhaustion.

Some specific points:

Fig 1 – It appears that low dose DAC did not increase proliferation (as the authors claim), as proliferation is not statistically different from untreated controls. What is the significance of increased CD3/CD25+ cells? Are these cells more activated? Are they Tregs? Increases in central memory and IL-2-secreting T cells fairly small...are these changes biologically meaningful? Statistics on whole methylome data should be included in the analysis.

Fig 2 – Why does tumor decrease over time in the NT condition after 48 hours (subfigure a)? Is data reliable after this time point? Further, is media exchanged over this time to ensure that tumor growth is not impacted by lack of extracellular nutrients? Please describe how cell index is quantified. It would be helpful to include the quantitative data shown in Extended Fig 2b within primary figure 2. For subfigure b, how many donors were used? What statistical method was used? Subfigures c and d are incorrectly cited in the manuscript text. Subfigure c appears to show data from an experiment that was not repeated? In general, the gating strategy for the FACS experiments is unclear. Are cell population frequencies representing CAR+ cells only, or mixed

CAR+ and non-transduced? If mixed, is it critical to include data that allows specific definition on CAR+ cells.

Fig 3 – Please indicate what “gene expression value” is (ex. TPM, FPKM, etc.)? Please include statistics for RNA-seq expression data. A schematic could help better clarify subfigure d...it’s difficult to understand your methodology here. Further, please indicate whether differences in expression pre-/post-activation are statistically significant.

Fig 4 – It appears that much of the data in this figure is shown from 1 experiment which was not repeated. Reproducibility is essential to establish sufficient rigor to validate the conclusions. Why aren’t CAR T cells detected in the blood at early timepoints? In subfigure c, please show the x axis for the histograms. At what timepoint were samples taken from mice for analyses of CD25, CD62L, and PD-1/TIM-3/LAG3? For subfigure g, how many paraffin sections per mouse were quantified for statistical analyses? In subfigure f, please address why significant differences in exhaustion marker expression limited to PD-1? What is the methylation status at the PD-1 locus? Histograms and MFI data of exhaustion markers would be helpful. Finally, please clarify from which model (Raji or Nalm6) were tissues/cells harvested in subfigures c-e, and g.

Extended Fig 3 – CD8s treated with DAC have a significant defect in proliferation...is this expected or even beneficial? How does this translate to superior proliferation in vivo? Please provide statistics for subfigure e.

Extended Data Fig 4 – Please include statistics for methylome data. I’m concerned by the variability between replicate samples, which requires that the work be repeated with additional donors in order to establish a stronger pattern discerning the methylomes of CAR and dCAR T cells. The order of subfigures b and c are switched in the legend.

Methods – Please include your methods for analyzing the 850k DNA methylome and RNA-seq data.

Reviewer #2 (Remarks to the Author): expertise in T cell activation and epigenetics

In the manuscript “Low-dose decitabine priming endows CAR T cells with enhanced and persistent anti-tumour potential via epigenetic reprogramming”, the authors demonstrated that CAR T cells briefly cultured in the presence of decitabine, an inhibitor of DNA methyl transferases, during expansion show stronger anti-tumour activity than CAR T cells in a xenogeneic humanized murine ALL and NHL models. The authors have previously published a number of works showing that low-dose decitabine therapy improves immunotherapy responses in vivo (Wang et al., JCO 2018; Chen et al., IJC 2018). In these trials, the authors have postulated that decitabine in part works by affecting the T cell response to tumours. However, decitabine also affects tumour cell intrinsic biology as well as the biology of almost all cells within the body. The direct effect of decitabine on T cells in an immunotherapy context is less clear. This work attempts to resolve this question, in addition to showing that decitabine may be used to improve the function of CAR T cells in adoptive immunotherapy. The findings are therefore of general importance. However, it remains unclear from this work how brief exposure to decitabine during expansion enables such a profoundly improved response to tumours on a mechanistic level and much more work would be required to justify the claims made that the basis for improved responses is dependent on any of the following: increased cytokine production, improved T cell survival/persistence, improved memory marker expression, less exhaustion marker expression.

In addition, the manuscript is difficult to read through, as there are multiple typos and grammatical errors throughout. We recommend that the authors consider receiving the help of a native English speaker to correct and edit the writing. In addition, main figures presented are not always referred to in the main text, for example, Figure 1i and figure 3d were not mentioned

anywhere in the main text. The overall quality of this manuscript, including main text, figures and descriptions in the figure legend would need to be improved before we can recommend publication.

Major revisions:

1. The mechanistic basis for improved responses is not clear from the data presented. In particular, it would be important for the authors to show which of the following are most relevant to the improved anti-tumour potential of decitabine-primed cells: increased cytokine production, improved T cell survival/persistence, improved memory marker expression, less exhaustion marker expression.

2. Some findings appear at odds with others – for example, the authors present data that CD25 expression is increased on treated cells, indicating increased activation. However these cells also show an increased memory phenotype. Could the authors perform and present multiparameter flow cytometry to examine which subphenotype of cells have the memory phenotype, versus the more functional phenotype versus the more activated phenotype, ie are all the cells all of these things, or are different subpopulations doing these things independently? The figures and data presentation would benefit from presentations of flow cytometry data and better quality analysis of phenotypic subpopulations with separate analysis of CD4 and CD8 T cells.

3. The authors could perform better bioinformatics analysis to globally correlate differences in global gene expression with global changes in DNA methylation at the gene level, in addition to showing examples as the authors have done.

4. The authors should confirm promoter methylation status in CpG islands of genes such as PD-1 and Tim3 using bisulfite sequencing.

5. Is the upregulation of CD25 and an increased ratio of CD4+ T cells related to the induction of Treg cells. Could the authors examine Foxp3 and CD127 expression in addition to CD4 and CD25 expression specifically on CD4+ T cells?

6. It is not clear exactly when decitabine was added on the schematic in Fig 1a. Could this be shown schematically?

7. The authors could perform a better characterisation of exhaustion and memory markers at serial timepoints during the in vitro expansion and in vivo treatment specifically on phenotypic subfractions by multiparameter flow cytometry, showing representative flow dot plots in addition to graphs of replicate measurements.

8. Figure 1i and figure 3d were not mentioned anywhere in the main text.

9. It is not possible to see the separate GFP signal on the merged GFP RFP overlay in Figure 2b. Could the authors separately show GFP and RFP signals. Could the authors quantify this data.

10. Line 207, "memory-associated CD62L" expression (Fig. 4f)". There is no CD62L staining can be found in figure 4f.

11. Extended figure 5, 6, and 7 support the findings that dCAR T cells have stronger anti-tumour effect and demonstrated that they can eradicate tumour during re-challenge. We suggest these figures to be placed as main figures.

12. The data represented by lines on Fig. 1b are indistinguishable. Bigger bullets or different colours, need to be used.

13. The data could benefit in general from more representative presentations of the raw underlying data, whether this be flow cytometry or RNA-Seq plots.

Minor points:

Figure 2h and line 132, wrong spelling of EOMES

Line 108, the description is for Figure 2d, not 2c

Line 112 The description is for Figure 2c, not 2d
Extended figure 7f and 6e-f are not supplied.

To Reviewer 1:

Dear Reviewer:

All authors thank you for your positive evaluation of our manuscript. The comments you raised were very helpful for our revision. We have thoroughly considered your comments and have revised our manuscript as follows. According to your comments, I have provided a point-by-point response as follows.

Your comments:

First and foremost, several major experiments that support the manuscript's overall conclusion appear to have been conducted only once. Many of these are outlined below in the specific points, but include critical in vivo experiments. These must be repeated and exhibit reproducibility.

Authors' response: Authors' response: We thank you for your detailed review. As you noted, the repeatability of the experiment is very important. In this study, the in vitro experiments used T cells from 3-5 biological donors to culture CAR T cells for independent experiments. Each independent experiment was carried out with at least 3 technical replicates. Each animal experiment was repeated twice, and T cells from different biological donors were used for each independent experiment. The transcriptional profiling and methylation results were from 3 biological donors. Although we mentioned the reproducibility of some experiments in the figure legend, the representation was not detailed and accurate, and the animal experiment results only showed representative results, leading to concerns about the repeatability of the results. In the revised manuscript, the specific method with the experimental replicates is described in detail in the figure legend to ensure its accuracy. All the legends in the manuscript have been carefully revised. Animal experiments with modified results were pooled from two independent experiments, and all the original data were submitted.

Your comments:

Second, several primary and extended figures lack statistical rigor. Examples include methyomic and transcriptomic analyses (ex. Figures 1i, 3a, 3d), which the authors use to support their claim of DAC-mediated CAR T reprogramming. Careful examination of heatmaps in Extended Fig. 4a-b display a concerning degree of variability. Additional replicate samples and statistics should be used

to fully validate the degree to which DAC induces cell reprogramming.

Authors' response: According to your comment, we have used two new biological samples to sequence and reanalyse the transcription profiles, and the results of different genes in the transcription profile are cluster analysed according to function. In the results, “**fig1.i, extended fig4.a and b**” were changed to “**Fig.2.e, Fig. 5a and b**” in our revised protocol.

Your comments:

Third, a large portion of the beginning of the manuscript is devoted to describing differential effects of DAC on CD4s versus CD8s. In fact, low dose DAC enhanced CD4 proliferation, but counterintuitively abrogated CD8 proliferation *ex vivo*. The mechanism by which this occurs should be addressed. Further, methylomic and transcriptomic differences between CD4s and CD8s, as well as those potentially observed *in vivo*, are not addressed. This is a critical factor to understanding the effects of DAC on CAR T cell function and deserves further interrogation.

Authors' response: Authors' response: We appreciate your comments. According to your comments, we added experiments to compare the proliferation of CD4-positive and CD8-positive cells after repeated antigen stimulation in CAR T and dCAR T cells *in vitro*. The results showed that although the proliferation of the CD8-positive dCAR T cells *ex vivo* was lower than that of the CAR T cell group, this relatively lower proliferation rate was changed after antigen stimulation. For both the CD4-positive and CD8-positive T cells, the amplification capacity was significantly higher in the dCAR T cell group than in the CAR T group under repeated antigen stimulation.

We added “**Although the proliferation of the CD8-positive dCAR T cells was lower than that of the CAR T cell group *ex vivo*, this relatively lower proliferation rate was changed after antigen stimulation. For both the CD4-positive and CD8-positive T cells, the amplification capacity was significantly higher in the dCAR T cell group than in the CAR T group under repeated antigen stimulation (Fig. 3h).**” at lines 120-124 on page 6 of our revised manuscript.

The mechanism of functional changes in CD4-positive or CD8-positive dCAR T cells was not further explored in this study. However, in future research, further analysis of the mechanism of action of CD4-positive or CD8-positive dCAR T cells will be performed.

Your comments:

Finally, authors make the argument that ex vivo DAC treatment reduces the capacity of CAR T cells to become exhausted in vivo. This claim is supported mainly by FACS data, indicating a decrease in the frequency of PD-1+ cells in dCAR T cells (but not LAG-3 or TIM-3). Why is PD-1 only affected by DAC pre-treatment? It's possible that the increased proliferation and enhanced functionality of dCAR T cells in vivo is independent of exhaustion and perhaps related to suppression enforced by the tumor microenvironment. Careful methylomic and/or transcriptomic analyses of in vivo dCAR T cells is necessary to determine whether these cells are truly resistant to T cell exhaustion.

Authors' response:

Based on your suggestions, we added in vivo experiments to further evaluate the differences in the functional and transcriptional profiles of dCAR T cells and CAR T cells in tumours. Unexpectedly, the biggest difference between the dCAR T cells in tumours is the increase in memory T cells, which also explains why for the tumour cell restimulation model, the dCAR T group showed significant CAR T cell re-expansion. However, regarding exhaustion, as you said, dCAR T's low exhaustion may be due to increased memory. The results of this experiment have been added to the revised manuscript.

To further assess phenotypic and genome-wide changes associated with anti-tumour function, we modified the Raji NHL model experimental conditions to delay tumour regression and obtained CAR T cells in the tumour site to perform phenotype testing and RNA-seq (Fig.9.a). The tumour sizes of the dCAR T cell group were similar to those of the CAR T cell group on day 7 after cell infusion but were smaller than those of the CAR T group on day 14 (Fig.9.b). The tumour infiltrating CAR T recoveries were higher in the dCAR T cell group on day 7 and/or day 14 after cell infusion than the CAR T group, and dCAR T cells showed significantly significant expansion and homing advantages (Fig.9.c). The Tcm markers and CD25 expression were significantly upregulated in the dCAR T cells, including CD8-positive and CD4-positive dCAR T cells, on the 7th and 14th days after cell infusion. Seven days after cell infusion (Fig.9.e and f), the CD8+PD-1+TIM3+cells were highly expressed in both the CAR T and dCAR T groups (Fig.9.d). The CD8-positive dCAR T cells showed a mild but statistically significant increase in PD-1/TIM3 expression compared to the CAR T cells, but after 14 days of

cell infusion, CD8+PD-1+TIM+ was strongly up-regulated in the CAR T cells and higher than that in dCAR T cells (Fig.9.d).

Notably, others have described highly infiltrated tumours with a distinct population of TIM3+ cells, which resemble phenotypically exhausted CD8 T cells¹⁷. As the tumour-infiltrating dCAR T cells showed an exhausted phenotype on day 7 after cell infusion, we performed RNA-seq on the tumour-infiltrating dCAR T and CAR T cells to further investigate the T cell function-associated gene status of the dCAR T cells and differences compared with the CAR T cells. Tumour-infiltrating dCAR T cells on day 7 showed increases in activation/exhaustion and proliferation-associated genes (Fig. 10.c). In contrast, these exhaustion genes were downregulated in the tumour-infiltrating dCAR T cells compared with the CAR T cells on day 14 after cell infusion (Fig. 10.c). Throughout, the tumour-infiltrating dCAR T cells had higher levels of memory-associated genes (Fig. 10.c). Differential Genes enrichment Go results revealed upregulation of the expression of memory-associated and cell proliferation-related GO items in the dCAR T cells compared with the CAR T cells on days 7 and 14 (Fig.10.b). Here, the dCAR T cells also exhibited stronger expansion and viability of the CD8-positive tumour-infiltrating dCAR T cells, which showed higher TCM and CD25 expression, than the CAR T cells (Fig.9.e and f); thus, the relatively low exhaustion status of the dCAR T cells in the later period may be because these cells contain more stem cell-like cells. To confirm that these gene transcriptome changes are due to changes in the methylation status after DAC treatment, we further analysed the changes in the transcription profiles of the dCAR T and CAR T cells from the resting state to infiltration to the tumour state. As under stimulation by tumour antigens, both the CAR T cells and the dCAR T cells showed major significant transcriptome changes. The dCAR T cells did not exhibit the same genetic changes as the CAR T cells (Fig. 10.d). These results suggest that the reprogramming effect on dCAR T cells is exerted and magnified after antigen exposure through treatment with DAC rather than by CAR T cell-intrinsic factors (Fig. 10.e).

Your comments:

Fig.1- It appears that low dose DAC did not increase proliferation (as the authors claim), as

proliferation is not statistically different from untreated controls. What is the significance of increased CD3/CD25+ cells? Are these cells more activated? Are they Tregs? Increases in central memory and IL-2-secreting T cells fairly small...are these changes biologically meaningful? Statistics on whole methylome data should be included in the analysis.

Authors' response:

As you noted, the effect of DAC on the proliferation of CAR T cells is not significant in the resting state. The substantial proliferation advantage of dCAR T cells is exhibited after antigen stimulation, especially in vivo. Our description was not accurate in the Results section; therefore,

“The 10-nM dose of DAC had little or no effect on CAR T cell viability, but increased the proliferative capacity during CAR T cell culture (Fig.1b).” to **“The 10-nM dose of DAC had little or no effect on CAR T cell viability and proliferative capacity during CAR T cell culture (Fig. 1b).”** at line 58-59 in page 3 of our revised manuscript.

During the study, we also assayed the contents of Tregs and Th17 cells in dCAR and CAR T cells. Because the population of these two cell types was relatively low and had no difference in the dCAR and CAR T cells, we have not shown these data in the manuscript. We have added this result **“the population of regulatory cells (Treg) and Th17 cells are very low and was not different in the dCAR and CAR T groups (Supplementary fig.1)”** at line 65-66 in page 4 of our revised manuscript to the manuscript.

CD3+CD25+ expression was slightly upregulated after DAC treatment but was negligible compared to that after antigen activation (CD3+CD25+ expression more than 50%); the slight increase here may be caused by the response to IL-2 added during cell culture. As you noted, the difference in cell phenotype, including Tcm and IL-2, seemed relatively weak in the dCAR T cells relative to the CAR T cells in the resting state. In subsequent antigen stimulation experiments and animal models, the differences in the central memory phenotype and cell proliferation became increasingly significant. Here, we examined the effects of DAC on CAR T cells in different states and proved that these effects were amplified after antigen stimulation. Finally, we added results from the whole methylome data analysis **“Methylation analysis. The Illumina Infinium Methylation**

EPIC (850K) BeadChip was used for epigenome-wide DNAm measurement. Chips were scanned on an Iscan (Illumina). Data was processed using Illumina Genome Studio as well as JMP 11 (SAS). ”to the “Methods and Materials” in the revised manuscript.

Your comments:

Fig 2 – Why does tumour decrease over time in the NT condition after 48 hours (subfigure a)? Is data reliable after this time point? Further, is media exchanged over this time to ensure that tumor growth is not impacted by lack of extracellular nutrients? Please describe how cell index is quantified. It would be helpful to include the quantitative data shown in Extended Fig 2b within primary figure 2. For subfigure b, how many donors were used? What statistical method was used? Subfigures c and d are incorrectly cited in the manuscript text. Subfigure c appears to show data from an experiment that was not repeated? In general, the gating strategy for the FACS experiments is unclear. Are cell population frequencies representing CAR⁺ cells only, or mixed CAR⁺ and non-transduced? If mixed, is it critical to include data that allows specific definition on CAR⁺ cells.

Authors’ response:

We repeated two independent continuous cytotoxicity assays with two donor-derived T cell samples, exhibited representative results and modified the method in detail in the revised manuscript. During the entire experiment, fresh medium was added once to ensure the nutrient composition around the cells, and the quantitative indicators are explained in detail. The sentences **“With the xCELLigence impedance-based system, continuous tumour cell killing was evaluated over 150 hours. Raji cells were plated in a 96-well, resistor-bottomed plate at 10,000 cells per well. After 18–24 hours, 2,000 effector T cells were added in triplicate, at which point cell index values correlating to Raji adherence were normalized. Impedance-based measurements of the normalized cell index were collected every 15 min and were determined by measurement of the impedance of a current across the transistor plate caused by tumour cell adherence.”** were added to replace **“The continuous cytotoxicity assays of CAR T cells were monitored using an xCELLigence Real-Time Cell Analyzer-Multiple Plate system (Agilent Technologies). This platform measures un-killed adhesive target cells in real time. First, Raji cells (10000), CD40-antibody and 100µl culture medium were plated in each well of a 96-well (E-Plate 96,**

Agilent Technologies, USA), which was then placed into the Real-Time Cell Analyzer-Multiple Plate instrument at 37 °C with 5% CO₂. After 24 hours, 2,000 effector T cells suspended in 100 µl culture media were added in triplicate. After 72 hours, 100 µl fresh culture medium were replaced. Cell index values were calculated using the RTCA Software. The cell index represented changes in electrical impedance, and reflected the number of un-killed target cells on biocompatible microelectrode surfaces. For real-time monitoring, the cell index was read automatically every 15 min. Cell index data in each group represented the mean value from three wells.” in the Methods section in the page 15 of our revised manuscript. According to your suggestion, the quantitative data of continuous cytotoxicity were added in Fig.3.a. Subfigures c and d have been corrected.

For subfigure b, we used a total of 3 donor-derived T cells to repeat this experiment. The conclusion of each independent experiment is consistent, and a representative result is shown here. To accurately display the results, we added the statistics of three independent experiments at 4 time points in the supplementary material.

The cell population frequencies representing CAR⁺ cells are shown in Subfigure c. In this study, to minimize the interference caused by non-CAR T cells in every experiment, including cell experiments and mouse experiments, we used CAR T cells, which were sorted three days after CAR infection. In the previous text, we did not clearly describe the source of the cells. We have supplemented the cell sorting process in the revised text. The sentence **“Purified CAR T cells were obtained by sorting (magnetic beads) at 3 days post-lentivirus infection.”** was added at lines 288-289 on page 14 of our revised Materials. For Subfigure c, the statistical method was added to the figure legend in the revised manuscript.

Your comments:

Fig 3 – Please indicate what “gene expression value” is (ex. TPM, FPKM, etc.)? Please include statistics for RNA-seq expression data. A schematic could help better clarify subfigure d...it’s difficult to understand your methodology here. Further, please indicate whether differences in expression pre-/post-activation are statistically significant.

Authors’ response:

The Fig. 3 "gene expression value" is FPKM, and the details including statistical data of the

genes were added to the revised manuscript. In order to better express the function of T cells, we reanalyzed the diagram and replaced it with a heat map. The result is shown in Fig.2.g.

Based on your suggestion, we have added schematics show that dCAR T cells behave differently from CAR T cells before and after antigen stimulation. The schematic diagram (Fig.10.e) is added to the end of the text. I cannot confirm whether this schematic diagram is completely in line with your requirement. If not, please let me know and give me more suggestion.

Fig.10.e

Your comments:

Fig 4 – It appears that much of the data in this figure is shown from 1 experiment which was not repeated. Reproducibility is essential to establish sufficient rigor to validate the conclusions. Why aren't CAR T cells detected in the blood at early timepoints? In subfigure c, please show the x axis for the histograms. At what timepoint were samples taken from mice for analyses of CD25, CD62L, and PD-1/TIM-3/LAG3? For subfigure g, how many paraffin sections per mouse were quantified for statistical analyses? In subfigure f, please address why significant differences in exhaustion marker expression limited to PD-1? What is the methylation status at the PD-1 locus? Histograms and MFI data of exhaustion markers would be helpful. Finally, please clarify from which model (Raji or Nalm6) were tissues/cells harvested in subfigures c-e, and g.

Authors' response:

We agree with this point. In our study, each mouse experiment was repeated twice, and T cells from different biological donors were used for each independent experiment. We apologize for the confusion caused by our unclear description. We have revised the statistical methods and results in

the new manuscript.

The results of clinical trials published by many institutions have shown that the peak of CAR T cell expansion in peripheral blood is approximately 11-17 days after cell infusion, so we began to detect CAR T cells in peripheral blood by FACS from day 7 to determine whether there is a difference in the peak of proliferation between the CAR T cell group and the dCAR T cell group in vivo. In fact, CAR T cells can be detected in peripheral blood on day 3 after cell infusion, but due to the concern that the CAR T amplification peak may not be reached in PB, FACS detection may introduce errors. Therefore, we used Q-PCR to detect the presence or absence of CAR T cells in PB, and the results **“the CAR gene number in the peripheral blood (PB) was obviously stronger in the dCAR T group than in CAR T group (Fig. 6.d).”** are shown in revised Fig.6.d and the Results section (page 9, lines 178-179).

We apologize for the incomplete data displayed in Fig. 4. According to your comments, we have improved the data in the revised manuscript, including the time of detection, the statistical method for paraffin section analysis and the source of the test specimen. **“Mice of Raji-bearing mice treated with 1×10^6 CAR T cells were sacrificed on day 14 after treatment. Paraffin sections of tumours were stained with CAR scFv-, Granzyme B-, IFN- γ - and TNF- α -specific probes for RNA ISH. Microscopic magnification: 10 \times and 40 \times . The data represent the mean \pm s.e.m. from 5 slices of 3 mice of two independent experiments, and the positive cells count value from the mean value of randomly 10 fields in each silce. P values were calculated with a two-tailed unpaired t test. *p < 0.05, **p < 0.01, ***p < 0.001, and ****p < 0.0001.”** were added in Figure legend of Fig.8.b. To clearly show the results of each animal experiment, we adjusted the order of the results in the revised manuscript and combined the results of the same experiment to increase the clarity.

Your comments:

Extended Fig 3 – CD8s treated with DAC have a significant defect in proliferation...is this expected or even beneficial? How does this translate to superior proliferation in vivo? Please provide statistics for subfigure e.

Authors' response:

Thank you for your valuable comments. We added another experiment to compare the

proliferation of CD8-positive cells after repeated antigen stimulation in CAR T and dCAR T cells in vitro. The results suggest that after antigen stimulation, the proliferative ability of the CD8-positive dCAR T cells is better than that of the CAR T cells, and the same results were also observed in animal experiments. The same relatively high expression was observed for proliferation-associated genes. Some studies have shown that improve the ratio of CD4 to CD8 is favourable for the effect of cell therapy, and DAC seems to increase the CD4 level in CAR T cells in vitro and enhance the proliferation and killing capacity of CD8-positive cells after antigen stimulation. Therefore, here, we think that it is beneficial to increase the proliferative capacity of CD4-positive cells in an ex vivo environment.

Your comments:

Extended Data Fig 4 – Please include statistics for methylome data. I'm concerned by the variability between replicate samples, which requires that the work be repeated with additional donors in order to establish a stronger pattern discerning the methylomes of CAR and dCAR T cells. The order of subfigures b and c are switched in the legend.

Authors' response:

Following your suggestion, we used two new donors to repeated chip-based 850k whole DNA methylome analysis. We readjusted the subgraphs c and d, and improved the statistics data of methylation in revised manuscript.

Your comments:

Methods – Please include your methods for analyzing the 850k DNA methylome and RNA-seq data.

Authors' response:

According to your helpful suggestion, we improved the statistics data of methylation and RNA-seq in revised manuscript. **“The RNA-seq data of gene difference analysis is the original readcount data obtained in gene quantification. First normalizes the original readcount, mainly to correct the sequencing depth. Then calculate the hypothesis test probability (p-value) through the statistical model, and finally perform multiple, repeated hypothesis test correction (BH) to get the FDR value (false discovery rate, padj is its common form, the following are all used padj to represent FDR) . DESeq2 software was used to analyse the significance of expression difference,**

and p_{adj} was less than 0.05 as the difference significance standard.” were added in section of the **“Materials and Methods”** of the revised manuscript.

To Reviewer 2:

Dear Reviewer:

All authors thank you for your positive evaluation of our manuscript and your helpful suggestions. Your comments were very informative. I appreciate your valuable comments and have carefully revised our manuscript as follows. According to your comments, I have provided a point-by-point revision as follows.

Your Major Comments:

1. The mechanistic basis for improved responses is not clear from the data presented. In particular, it would be important for the authors to show which of the following are most relevant to the improved anti-tumour potential of decitabine-primed cells: increased cytokine production, improved T cell survival/persistence, improved memory marker expression, less exhaustion marker expression.

Authors' response:

Thank you for your valuable comments. Indeed, we did not determine which antitumour potentials of dCAR T cells are most relevant to decitabine treatment. According to the suggestions of you and another reviewer, we have added an intratumoural CAR T experiment based on a mouse model to try to determine whether there are differences in the phenotype or gene expression of dCAR T cells and CAR T cells under long-term stimulation of tumour antigens *in vivo*. Unlike the *in vitro* experiments, TIL-CAR T cells showed another differentiation process. On the 7th and 14th days after treatment, TIL-dCAR T cells showed rapid proliferation and activation, and unexpectedly, dCAR T's memory phenotype and memory-associated genes showed a significant increase in every status. Based on these results, we further analysed the methylation level of memory-related genes in dCAR T cells in the resting state and after antigen stimulation *in vitro* and found that the methylation of memory-related genes was absent in both statuses. Therefore, we speculate that the persistent memory improvement of CAR T cells may be the tumour-related activity that is most relevant to treatment with decitabine. Experiments related to DAC mechanisms will continue in the future in our research laboratory.

The results of the intratumoural experiment were carefully added to separate sections in the revised manuscript.

Results: To further assess phenotypic and genome-wide changes associated with anti-tumour function, we modified the Raji NHL model experimental conditions to delay tumour regression and obtained CAR T cells in the tumour site to perform phenotype testing and RNA-seq (Fig.9.a). The tumour sizes of the dCAR T cell group were similar to those of the CAR T cell group on day 7 after cell infusion but were smaller than those of the CAR T group on day 14 (Fig.9.b). The tumour infiltrating CAR T recoveries were higher in the dCAR T cell group on day 7 and/or day 14 after cell infusion than the CAR T group, and dCAR T cells showed significantly significant expansion and homing advantages (Fig.9.c). The Tcm markers and CD25 expression were significantly upregulated in the dCAR T cells, including CD8-positive and CD4-positive dCAR T cells, on the 7th and 14th days after cell infusion. Seven days after cell infusion (Fig.9.e and f), the CD8+PD-1+TIM3+cells were highly expressed in both the CAR T and dCAR T groups (Fig.9.d). The CD8-positive dCAR T cells showed a mild but statistically significant increase in PD-1/TIM3 expression compared to the CAR T cells, but after 14 days of cell infusion, CD8+PD-1+TIM+ was strongly up-regulated in the CAR T cells and higher than that in dCAR T cells (Fig.9.d).

Notably, others have described highly infiltrated tumours with a distinct population of TIM3+ cells, which resemble phenotypically exhausted CD8 T cells¹⁷. As the tumour-infiltrating dCAR T cells showed an exhausted phenotype on day 7 after cell infusion, we performed RNA-seq on the tumour-infiltrating dCAR T and CAR T cells to further investigate the T cell function-associated gene status of the dCAR T cells and differences compared with the CAR T cells. Tumour-infiltrating dCAR T cells on day 7 showed increases in activation/exhaustion and proliferation-associated genes (Fig. 10.c). In contrast, these exhaustion genes were

downregulated in the tumour-infiltrating dCAR T cells compared with the CAR T cells on day 14 after cell infusion (Fig. 10.c). Throughout, the tumour-infiltrating dCAR T cells had higher levels of memory-associated genes (Fig. 10.c). Differential Genes enrichment Go results revealed upregulation of the expression of memory-associated and cell proliferation-related GO items in the dCAR T cells compared with the CAR T cells on days 7 and 14 (Fig.10.b). Here, the dCAR T cells also exhibited stronger expansion and viability of the CD8-positive tumour-infiltrating dCAR T cells, which showed higher TCM and CD25 expression, than the CAR T cells (Fig.9.e and f); thus, the relatively low exhaustion status of the dCAR T cells in the later period may be because these cells contain more stem cell-like cells. To confirm that these gene transcriptome changes are due to changes in the methylation status after DAC treatment, we further analysed the changes in the transcription profiles of the dCAR T and CAR T cells from the resting state to infiltration to the tumour state. As under stimulation by tumour antigens, both the CAR T cells and the dCAR T cells showed major significant transcriptome changes. The dCAR T cells did not exhibit the same genetic changes as the CAR T cells (Fig. 10.d). These results suggest that the reprogramming effect on dCAR T cells is exerted and magnified after antigen exposure through treatment with DAC rather than by CAR T cell-intrinsic factors (Fig. 10.e).

2. Some findings appear at odds with others – for example, the authors present data that CD25 expression is increased on treated cells, indicating increased activation. However, these cells also show an increased memory phenotype. Could the authors perform and present multiparameter flow cytometry to examine which subphenotype of cells have the memory phenotype, versus the more functional phenotype versus the more activated phenotype, i.e. are all the cells all of these things, or are different subpopulations doing these things independently? The figures and data presentation

would benefit from presentations of flow cytometry data and better quality analysis of phenotypic subpopulations with separate analysis of CD4 and CD8 T cells.

Authors' response:

Thank you for your comments. According to your suggestion, especially in the supplemented intratumour experiment. The tumour infiltrating CAR T cells were classified as CD4- and CD8-positive cell types for further analysis. The result has been added to the modified document.

3. The authors could perform better bioinformatics analysis to globally correlate differences in global gene expression with global changes in DNA methylation at the gene level, in addition to showing examples as the authors have done.

Authors' response:

Thank you for your valuable comments. Indeed, we did not further analyse the correlation between gene transcription and methylation level. According to your suggestion, we performed methylation correlation analysis with corresponding transcriptional changes before and after antigen stimulation. Epigenetic changes occurred near key genes involved in memory, TCF7, CD25 and many other checkpoint CTLA4 molecules (**Fig. 2.de, Fig.5.f**), suggesting changes in these genes due to demethylation. These results were added to the revised manuscript. **“The differential genes in the transcription profile also have co-ordinately change methylation differential sites exist, especially in memory-associated genes such as BCL6, TCF7, LEF, and ILR7 and exhaustion/aging-associated genes such as AKT3, EOMES, NR4A3 and CTLA4 (fig.2.d).”** were added at lines 93-96 on page 5, **“There were consistency differential methylation CpG sites in these genes (Fig.5.f).”** were added at lines 152 on page 8 of our revised manuscript.

4. The authors should confirm promoter methylation status in CpG islands of genes such as PD-1 and Tim3 using bisulfite sequencing.

Authors' response: According to your comment, we used bisulfite sequencing to confirm the methylation status of the promoter in CpG islands of PD-1 and TIM3 gene. The results showed that Tim3 and PD-1 was partially methylated in day 7 effector cells. Moreover, treatment of the activated

T cells with DAC resulted in complete demethylation of CpG sites in Tim3 and PD-1, this was confirmed that PD-1 and TIM3 genes were regulated by methylation control. The results "**To obtain nucleotide resolution of DNA methylation, we performed bisulfite sequencing of Tim3 and PD-1 on genomic DNA from in vitro activated primary T cells cultured in the presence and absence of the DAC. Tim3 and PD-1 was partially methylated in day 7 effector cells. Moreover, treatment of the activated T cells with DAC resulted in complete demethylation of CpG sites in Tim3 and PD-1.**" were added at lines 205-209 on page 10 of our revised manuscript. And the bisulfite sequencing associated method "**Bisulfite modification, methylation-specific PCR (MSP) and bisulfite sequencing DNA was prepared by the proteinase K method. Bisulfite treatment was carried out as previously described.1,2 MSP primers were designed according to genomic sequences around transcription start sites (TSS) and synthesized to detect unmethylated and methylated alleles. Bisulfite sequencing (BSSQ) was performed as previously described.3 BSSQ products were amplified by primers flanking the targeted regions and included MSP products.**" were added in "Materials and Methods" revised manuscript.

5. Is the upregulation of CD25 and an increased ratio of CD4+ T cells related to the induction of Treg cells. Could the authors examine Foxp3 and CD127 expression in addition to CD4 and CD25 expression specifically on CD4+ T cells?

Authors' response: According to your helpful suggestion, we also assayed the contents of Treg (CD3+CD4+CD25+FoxP3+) and Th17 (CD3+CD8+IL-17+) cells in the dCAR and CAR T cells. Because the population of these two cell types was relatively low and had no difference in dCAR and CAR T cells, we have not shown these data in the manuscript. Based on your comments, we added the sentence "**the population of Treg and Th17 cells are very low and have no difference in dCAR or CAR T group (Supplement fig2)**" at lines 65-66 on page 4 of our revised manuscript.

6. It is not clear exactly when decitabine was added on the schematic in Fig 1a. Could this be shown schematically?

Authors' response: We added DAC at the same time as the CAR gene infection and removed DAC after 24 hours. According to your suggestion, we have redrawn the schematic diagram and marked the detail in figure legend. The sentence “**a, The flow chart of dCAR T cell-culture and functional test. DAC was added at the same time as the CAR gene infection and removed after 24 hours.**” was added in **Fig.1. legend of** revised manuscript.

7. The authors could perform a better characterisation of exhaustion and memory markers at serial timepoints during the in vitro expansion and in vivo treatment specifically on phenotypic subfractions by multiparameter flow cytometry, showing representative flow dot plots in addition to graphs of replicate measurements.

Authors' response:

All authors thank you for your detailed review. According to your suggestion, we detected tumour infiltrating CD4 and CD8 CAR T cells in the added mouse experiments at several time points and determined that the expression of TCM and CD25 in both the CD4-positive and CD8-positive cells of the dCAR T group was significantly higher than that in the CAR T group. The expression of CD8+PD+TIM3+ in the dCAR T group was higher than that in the CAR T group on day 7 and tended to be stable and lower than that in the CAR T group on the 14th day. The results were added to the revised manuscript.

8. Figure 1i and figure 3d were not mentioned anywhere in the main text.

Authors' response: We apologize for our mistakes. We have carefully revised the correspondence between the figures and the results in the manuscript and corrected the errors in the revised manuscript.

9. It is not possible to see the separate GFP signal on the merged GFP RFP overlay in Figure 2b. Could the authors separately show GFP and RFP signals. Could the authors quantify this data.

Authors' response:

The images were collected in red and green channel separately, and merged automatically by Incucyte

Analysis Software. So the Images showed in manuscript were merged. The positive (red or green in color) cells were counted automatically as well. The experiment was repeated at least 3 times with similar results. Sorry there is no way to separate the images that have been merged according to the fluorescent color. If you think this separate imaging is necessary, please let us know and we can try our best to repeat the experiment according to your requirements.

10. Line 207, “memory-associated CD62L” expression (Fig. 4f)”. There is no CD62L staining can be found in figure 4f.

Authors’ response:

We mistakenly marked 4e as 4f. We apologize for this error and have corrected the sentence in the revised manuscript.

11. Extended figure 5, 6, and 7 support the findings that dCAR T cells have stronger anti-tumour effect and demonstrated that they can eradicate tumour during re-challenge. We suggest these figures to be placed as main figures.

Authors’ response:

All authors agree with your point. In the revised manuscript, we partially adjusted the order of Extended Figures 5, 6, 7 and 4, and used these 4 figures as the main figures.

12. The data represented by lines on Fig. 1b are indistinguishable. Bigger bullets or different colours, need to be used.

Authors’ response: As you mentioned, we readjusted the colours to highlight the differences in Fig.1b in the revised manuscript.

13. The data could benefit in general from more representative presentations of the raw underlying data, whether this be flow cytometry or RNA-Seq plots.

Authors’ response: Based on your suggestions, we have redrawn the methylation heat map and RNA-seq heat map in the revised manuscript. “Fig.3 and Fig.5” were revised.

Minor points:

1. Figure 2h and line 132, wrong spelling of EOMES
2. Line 108, the description is for Figure 2d, not 2c
3. Line 112 The description is for Figure 2c, not 2d
4. Extended figure 7f and 6e-f are not supplied.

Authors' response: Thank you for noting these errors. According to your comments, we have fixed these errors in the revised manuscript.

REVIEWER COMMENTS

Reviewer #1 (Remarks to the Author):

This revised manuscript is improved compared to the first submission. Authors reanalyzed methylomic and transcriptomic samples, conducted phenotypic and functional analyses on purified CD4 and CD8 CAR T cells both before and after antigen stimulation, conducted RNA- and bi-sulfite sequencing on tumor-infiltrating CAR T cells, and added more detail to the figure legends to clarify their experimental and statistical methodology. While many of our comments were adequately addressed, there are still significant issues with the manuscript:

Major

- While the authors have noted that 2 new biological samples were used for methylomic and transcriptomic analyses in Figure 2, it appears that the two samples used for these analyses in the first submission were omitted. Authors should justify why these samples were omitted. It's concerning that a such a critical figure to the study's conclusions is underpowered with only n=2 samples per group.
- There are many instances in which authors cite figures out of order. Please ensure that figures are cited in the order in which they're presented, as it is difficult for the reader to follow (ex. some subfigures from Fig. 4 are cited in the text before Fig. 2 and 3).
- Figures 3 and 4 could be consolidated, and 9 and 10 could be consolidated (with several subfigures moving to the data supplement).
- The authors demonstrate convincing evidence that 1) dCAR T cells exhibit a proliferative advantage in vitro and in vivo and 2) dCAR T cells exhibit a more memory-like phenotype, even in the face of continuous antigen stimulation. These findings are both interesting and important to the field of adoptive cell therapy. However, authors also go to great lengths to claim that dCAR T cells avoid T cell exhaustion, for which there is scant evidence. In vitro assays in which CAR T cells are stimulated 3x with tumor do not demonstrate antigen-induced dysfunction over time...the defining hallmark of T cell exhaustion. Moreover, phenotypically these cells do not resemble those that are exhausted, as only 6% of control CAR T cells express PD-1. Figure 5 claims that dCAR T cells exhibit lower expression of exhaustion-associated markers...however, these cells were only stimulated for 24 hours prior to analysis, which is not sufficient to induce exhaustion. The authors' in vivo models offer the most physiologic setting in which to observe exhaustion; however, again, the data shown does not convincingly demonstrate an exhausted phenotype in control CAR T cells both in terms of surface phenotype (Figure 8) or transcriptome (Figure 10). The conclusion could be strengthened by the use of GSEA to compare the in vivo RNA-seq data sets generated here with previously published sets from exhausted TILs or LCMV-specific exhausted T cells.
- In several places in the manuscript, the verbage is not clear and contradictory:
 - o For example, starting on line 119.... The manuscript discusses proliferation and simultaneously states that proliferative rates are higher, but also lower..."Across multiple donors, we consistently observed that dCAR T cells had significantly higher proliferation rates upon the second and third rounds of antigen stimulation (Fig.3.h). Although the proliferation of the CD8-positive dCAR T cells was lower than of the CAR T cell group in the ex vivo, this relatively lower proliferation rate was changed after antigen stimulated. For both the CD4 positive and CD8 positive T cells, the amplification capacity was significantly higher in the dCAR T cell group than in the CAR T group under repeated antigen stimulation (Fig.4.d)."
 - o Beginning on line 133, the meaning or context of the following statement is not clear, "Notably, the differentiation and depletion of T cells is antigen dependent"
- The volcano plots shown in Figure 10 display data unlike it is standard to visualize such plots. A statistical reviewer to analyze the data analysis would be of use.

Minor

- There are several typos throughout the text, please ensure that these are corrected.
- There doesn't appear to be any chronic antigen-stimulation experiments in Figure 1 despite the

schematic showing otherwise...these experiments, instead, appear in Figures 3 and 4. Please modify the schematic appropriately.

- On line 57, please mention the co-stimulatory domain on the CD19 CAR tested.
- Label the X-axis on Fig 1B.
- Is short-term, low-dose decitabine treatment known to induce stable DNMT3a degradation (Fig. 1f) in other published models? If so, please cite the paper(s).
- It might be more appropriate to include Figure 1g within Figure 2.
- For Figure 2d, are the differences in mean beta values between CART and dCART at these gene loci statistically significant? Please indicate on the graph and/or in the figure legend.
- In Figure 3, authors argue that the superior proliferation and maintenance of a memory-like phenotype of dCAR T cells after repeated (3x) antigen exposure occurs via prevention of T cell exhaustion. While I agree that these particular observations are interesting and important, there is not enough evidence to claim that these benefits occur by mitigating exhaustion (at least, at this point in the paper). To claim this, authors would need to show that 1) CAR T cells exhibit a decline in anti-tumor functionality after 3x antigen stimulation and 2) decitabine treatment prevents this dysfunction. I would advise authors to change the title of the figure to something along the lines of, "dCAR T cells exhibit enhanced anti-tumor reactivity and maintenance of a memory-like phenotype at low effector:target ratios", but mention in the manuscript text that these data suggest that prevention of exhaustion could be one mechanism by which decitabine treatment enhances anti-tumor functionality.
- The figure legend for 3d does not match the figure (which is a heatmap). It reads, "Bars show mean values with data points for 3 independent experiments. Experiments were repeated with four donor-derived T cells (n = 4). A two-tailed, paired, two-sample t test was used for statistical analysis. * P < 0.05, **P < 0.01, ***P < 0.001." Are these heatmaps mean values of 4 donors? Are there supposed to be bar graphs here instead of heatmaps?
- Please specify the difference between the two plots shown in Figure 4e. I assume the left plot was obtained from CD4s and the right from CD8s?
- Figures 6f and 7i show histograms that are intended to demonstrate an increase in the percentage of CAR+ cells in the blood of mice treated with dCAR T cells. It would be nice if these data were quantified (either embedded in the histograms themselves) or represented in bar graph form.
- Figure 9b – What statistical test was used here? Given the size of the error bars between the two groups being compared, I can't imagine there is statistical significance.
- Figure 9e – Please change the Y-axis to include the specific markers used assess Tcm phenotype (CD62L+ CD45RO+).
- Figure 10 – Please indicate how many samples were analyzed via RNA-seq in the figure legend.
- Figure 10a – Does this heatmap represent mean expression values from multiple samples for each condition/timepoint? Please specify.
- Figure 10b – Please use arrows to designate data points corresponding to the gene labels.
- Figure 10c – How were genes selected for each of these modules (memory, exhaustion/activation, proliferation, death/aging)? Was it biased or unbiased selection? Are there statistically significant differences?
- Subfigures B and C in Supplementary Figure 2 do not match their descriptions in the figure legend, please correct this.
- Please include the number of donors/samples for which bisulfite sequencing was conducted in Supplementary Figure 6.

Reviewer #2 (Remarks to the Author):

In the revised manuscript, the authors have made substantial changes to respond to our concerns. The revised manuscript is examines a potentially important means to optimise CAR T cell therapy and is of interest to the field. However, the current manuscript is quite difficult to read and still has a number of grammatical and syntactic errors, typos and inconsistency in the figure characters and

numbering throughout the text. In addition to the general concern in writing, there are several more issues that require the authors attention. These would need addressing before the manuscript is acceptable for publication

- In the new figure 9f, the authors showed the frequency of CD25+ / CD3+ T cells is higher in dCART without differentiating CD4+ from CD8+ in this analysis. Activation of these cells could be more accurately shown by using activation markers such as CD44 and CD69, in addition to CD25 in this in vivo experiment. Could the authors analyse on the RNAseq data of this experiment to support the activation state of CD4+ and CD8+ tumour infiltrating CAR and dCART.
- As mentioned in the text that CD8 dCAR T cells showed stronger expansion and viability than CAR T cells (line 246-247). However, a mixed CD4+/CD8+ result was presented in the figure (figure 9), which doesn't support the statement in the main text.
- The authors used a lot of "exhaustion markers" measuring exhaustion phenotype for both continues/repetitive antigen stimulations (figure3g-i) and single tumour inoculation (figure 9). The markers, such as PD-1, can be up-regulated upon activation without the cells being "exhausted". As such, while agreeing with that dCART cells can be less exhausted under repetitive antigen stimulation, the evidence doesn't support that tumour-infiltrating dCAR T cells are less "exhausted" in a short-term single tumour inoculation. Could the authors show or discuss on cytokine expression/secretion with RNAseq results to support the interpretation.
- Fig 1d: the authors use the term 'resting CAR T cells'. Are these cells resting or untreated with DAC?
- Fig 1f: how is the expression of CAR verified?
- Fig 2: The figures require higher resolution as it is not possible to read any of the legends.
- Fig 3i: Correct EOMES.
- Fig 4c: Raji model was used but the experiment set up was not mentioned properly in the text.
- Fig 4d: It is described in the text that both CD4 and CD8 positive dCAR T cells show higher proliferation than CAR T cells. This statement is not valid for CD8+ cells in figure 4d.
- Fig 2c is inappropriately referred in line 105
- Could the authors ensure that the figures are referred correctly in the main text, as well as a more readable and correct sentence structure in the manuscript.

Reviewer #3 (Remarks to the Author):

Comments for RNA-seq analysis:

1. The pre-processing of RNA-seq data (such as the sequence alignment) seems missed in the revised version of the manuscript.
2. The RNA-seq analysis description need to specify the normalization method they used in the data analysis.
3. The version of the DESeq2 package need to be specified.
4. "Then calculate the hypothesis test probability (p-value) through the statistical model, and finally perform multiple, repeated hypothesis test correction (BH) to get the FDR value (false

discovery rate, padj is its common form, the following are all used padj to represent FDR) . DESeq2 software was used to analyse the significance of expression difference, and padj was less than 0.05 as the difference significance standard.” Flows better if revised to “The DESeq2 package version xxx in the statistical analysis software R/Bioconductor was used to analyze the normalized RNA-seq data. Generalized linear models with empirical Bayes approach were used to identify significantly differentially expressed genes. The Benjamini-Hochberg (BH) method was used to adjust the raw P-values to control for the false discovery rate (FDR) in the data analysis. An adjusted P-value of 0.05 or less was regarded as significant difference.”

To Reviewer 1:

Dear Reviewer:

All authors express great thanks for your positive evaluation of our manuscript and your elaborative revision suggestions. Your comments are very instructive and are very helpful for us. I would like to fully accept your valuable comments and have carefully revised our manuscript as follows. According to your comments, I have given a point-by-point revised as follows.

Your major comments:

- While the authors have noted that 2 new biological samples were used for methylomic and transcriptomic analyses in Figure 2, it appears that the two samples used for these analyses in the first submission were omitted. Authors should justify why these samples were omitted. It's concerning that a such a critical figure to the study's conclusions is underpowered with only n=2 samples per group.

Authors' response:

We thank you for your detailed review. In the last manuscript, based on your comments and those of another reviewer and editor, we performed additional in vivo experiments to further evaluate differences in the phenotypes and transcriptomes of tumour-infiltrating dCAR T cells, which demonstrated a possible mechanism by which dCAR T cells exert their antitumour effects. We also performed analysis of the methylation and transcriptional profiles of the CAR and dCAR T cell samples used to supplement in vivo experiments, and these results are basically the same as those reported in the first version of the manuscript. To maintain the use of a consistent source of experimental samples from the cytology experiment to the final animal experiment, we replaced previous omics results with the results from the two samples that were used in the in vivo experiments in the last manuscript. We apologize for not clearly explaining why we omitted the omics results obtained with the first set of samples. As you said, research results obtained from two samples are not sufficient to support the conclusions of the whole article. We integrated the omics results from the first version of the manuscript into the second revised manuscript. The sentences **“This result was also observed in the expanded T cell samples, although to a lesser extent, as the transcription profiles results of 3 donors showed the upregulated expression of memory- and proliferation-associated genes and enhanced downregulation of the expression of T cell**

inhibitor-, death- and activation/exhaustion-related genes expression in the dCAR T cells compared to the CAR T cells (Supplemental Fig. 2) .” and “ These results also were observed in extended biological RNA-seq (Supplemental Fig. 5)” were added in the revised manuscript. (lines 92-96, page 5, lines 177-178, page 9).

- There are many instances in which authors cite figures out of order. Please ensure that figures are cited in the order in which they’re presented, as it is difficult for the reader to follow (ex. some subfigures from Fig. 4 are cited in the text before Fig. 2 and 3).

Authors’ response:

Based on your comment, we re-adjusted the order of the figures, especially for **Figs. 1 and 4**, to ensure that the figures are presented in an order consistent with that of the results in the article.

- Figures 3 and 4 could be consolidated, and 9 and 10 could be consolidated (with several subfigures moving to the data supplement).

Authors’ response:

Thank you for your suggestion. Combined with your last comment and to ensure that our readers better the results, we re-integrated the results in the figure with the legend “Enhanced cell functions in CD4- and CD8-positive CAR T cells after DAC treatment” into the revised manuscript, integrated the data from CD4- and CD8-positive dCAR T cells in the first paragraph into the fourth paragraph, and adjusted the order of the figures such that they are consistent with the order of the results presented in the fourth paragraph.

Based on your comment, **Figs. 9 and 10** were consolidated to **Fig. 9** in the revised manuscript. (pages 42 and 43)

- The authors demonstrate convincing evidence that 1) dCAR T cells exhibit a proliferative advantage in vitro and in vivo and 2) dCAR T cells exhibit a more memory-like phenotype, even in the face of continuous antigen stimulation. These findings are both interesting and important to the field of adoptive cell therapy. However, authors also go to great lengths to claim that dCAR T cells avoid T cell exhaustion, for which there is scant evidence. In vitro assays in which CAR T cells are stimulated 3x with tumor do not demonstrate antigen-induced dysfunction over time...the defining

hallmark of T cell exhaustion. Moreover, phenotypically these cells do not resemble those that are exhausted, as only 6% of control CAR T cells express PD-1. Figure 5 claims that dCAR T cells exhibit lower expression of exhaustion-associated markers...however, these cells were only stimulated for 24 hours prior to analysis, which is not sufficient to induce exhaustion. The authors' in vivo models offer the most physiologic setting in which to observe exhaustion; however, again, the data shown does not convincingly demonstrate an exhausted phenotype in control CAR T cells both in terms of surface phenotype (Figure 8) or transcriptome (Figure 10). The conclusion could be strengthened by the use of GSEA to compare the in vivo RNA-seq data sets generated here with previously published sets from exhausted TILs or LCMV-specific exhausted T cells.

Authors' response:

We appreciate your perceptive comments. Based on your comments, first, we performed additional experiments to compare the cytokine release functions of dCAR T and CAR T cells after repeated antigen stimulation for an extended period of time in vitro. The results showed that two weeks after constant stimulation by CD19-positive tumour cells, compared to the CAR T cells, the dCAR T cells exhibited a lower proportion of PD1- and TIM3-positive cells and produced higher levels of TNF α and IFN γ upon re-stimulation with Raji cells. The text **“These results were confirmed by further extension of the tumour cell stimulation time. After two weeks of constant stimulation by low-dose Raji cells, CAR T cells exhibited higher proportions of PD1+TIM3+ cells than dCAR T cells (Supplemental Fig. 4a). Compared to the CAR T cells, the dCAR T cells produced higher levels of IL-2, TNF- α and IFN- γ upon re-stimulation with Raji cells (Supplemental Fig. 4b).”** was added to the revised manuscript. (lines 126-130, pages 6-7)

Second, as you noted, in vitro experiments cannot recreate a complete physiological environment. Based on your suggestions, we used GSEA to compare the in vivo RNA-seq data with previously published datasets from tumour-infiltrating CAR T cells. To ensure that the results were as comparable as possible, we selected published data as consistent as possible with our study conditions for comparison (*Sonia Guedan, Aviv Madar, Victoria Casado-Medrano, et al. Single residue in CD28-costimulated CAR T cells limits long-term persistence and antitumour durability. J Clin Invest. 2020 Jun 1;130(6):3087-3097.*), the sample source and tumour-infiltrating CAR T cell collection time point in this paper were similar to our study. The data in this paper were collected from an NSG mouse model of human tumours in which CD19 CAR T cells were targeted, and the

tumour-infiltrating CAR T cells were collected on days 7 and 14 after cell transfer and used to perform RNA-seq. The GSEA analysis showed that the tumour-infiltrating CAR T cells on day 14 after cell infusion had similar and no significant difference gene enrichment in T cell cytokine secretion, T cell activation and memory compared to other previous reported tumour-infiltrating CAR T cells (Fig). These results suggested the tumour-infiltrating CAR T cells on day 14 in this study shown a similar dysfunction state as other report.

Fig. Representative GSEA enrichment plot, demonstrating the no-significant and similar expression of memory-, cytokine-, exhaustion- and programming death-related genes in tumor infiltrating CAR T cells versus tumor infiltrating CAR T-iso cells on 14 days after cell infusion.

- In several places in the manuscript, the verbage is not clear and contradictory:
- For example, starting on line 119.... The manuscript discusses proliferation and simultaneously states that proliferative rates are higher, but also lower....”Across multiple donors, we consistently observed that dCAR T cells had significantly higher proliferation rates upon the second and third rounds of antigen stimulation (Fig.3.h). Although the proliferation of the CD8-positive dCAR T cells was lower than of the CAR T cell group in the ex vivo, this relatively lower proliferation rate was changed after antigen stimulated. For both the CD4 positive and CD8 positive T cells, the amplification capacity was significantly higher in the dCAR T cell group than in the CAR T group under repeated antigen stimulation (Fig.4.d).”

Authors' response:

Thank you very much for your meticulous review. The text “Although the proliferation of the CD8-positive dCAR T cells was lower than of the CAR T cell group in the ex vivo, this relatively lower proliferation rate was changed after antigen stimulated. For both the CD4 positive and CD8 positive T cells, the amplification capacity was significantly higher in the dCAR T cell group than in the CAR T group under repeated antigen stimulation (Fig.4.d).” was revised to **“Furthermore, the amplification capacity of both the CD4-positive and CD8-positive T cells was significantly higher in the dCAR T cell group than that of in the CAR T cell group under repeated antigen stimulation (Fig. 4e).”** (lines 141-143, page 7).

- Beginning on line 133, the meaning or context of the following statement is not clear, “Notably, the differentiation and depletion of T cells is antigen dependent”

Authors' response:

Based on your suggestion, “Notably, the differentiation and depletion of T cells is antigen dependent,” was deleted from the original version of the manuscript.

- The volcano plots shown in Figure 10 display data unlike it is standard to visualize such plots. A statistical reviewer to analyze the data analysis would be of use.

Authors' response:

Based on your comments, we analysed published articles and modified the Volcano plot. Using R package software, statistical analysis of differential gene expression between CAR T and dCAR T cells was performed, and the resulting log₂ (fold change) and log₁₀ (p value) values were used to draw the Volcano plot. In the revised manuscript, the classification and number of differentially expressed genes (both upregulated and downregulated) are indicated in the figure, and the value calculated from differential expression analysis is included in the legend. The sentence “Volcano map of differentially expressed genes and enrichment in GO items among dCAR T and CAR T cells on days 7 and 14 after cell infusion. p < 0.05.” in the figure legend was revised to **“h, The Volcano plot illustrates fold changes in gene expression in tumour-infiltrating dCAR T cells compared with tumour-infiltrating CAR T cells. Upregulated genes were shown in red, downregulated genes were shown in blue, and genes whose difference in expression was statistically**

nonsignificant were shown in grey. Values were presented as the log₂ of the tag counts. Statistically significant was presented as $p < 0.05$.” (lines 850-854, page 43).

Xun Huang, Juan Yan 2, Min Zhang, et.al. Targeting Epigenetic Crosstalk as a Therapeutic Strategy for EZH2-Aberrant Solid Tumors. Cell. 2018 Sep 20;175(1):186-199.e19.

Fubing Li, Yang Li, Huichun Liang, et.al. HECTD3 mediates TRAF3 polyubiquitination and type I interferon induction during bacterial infection. J Clin Invest. 2018 Aug 31;128(9):4148-4162.

Your minor comments:

- There are several typos throughout the text, please ensure that these are corrected.

Authors’ response:

We acknowledge the language issues with the manuscript. To address this important issue, we have asked native English-speaking friends to edit the entire manuscript.

- There doesn’t appear to be any chronic antigen-stimulation experiments in Figure 1 despite the schematic showing otherwise...these experiments, instead, appear in Figures 3 and 4. Please modify the schematic appropriately.

Authors’ response:

The schematic in Fig.1 was changed per your suggestion in our revised manuscript.

- On line 57, please mention the co-stimulatory domain on the CD19 CAR tested.

Authors’ response:

The text, “**CAR (CAR-CD19-expressing) T cells were successfully prepared by transducing human peripheral blood mononuclear cells (PBMCs) with a lentivirus encoding an anti-CD19 scFV and 4-1BB/CD3 ζ CAR.**” was added to our revised manuscript (lines 54-56, page 3).

- Label the X-axis on Fig 1B.

Authors’ response:

The Label of X-axis on **Fig. 1b** was added in our revised manuscript.

- Is short-term, low-dose decitabine treatment known to induce stable DNMT3a degradation (Fig. 1f) in other published models? If so, please cite the paper(s).

Authors' response:

There have low-dose decitabine treatment in vivo could induce DNMT3a degradation, but no reports on the use of decitabine at similar doses to induce the degradation of DNMT3a in T cells in vitro.

- It might be more appropriate to include Figure 1g within Figure 2.

Authors' response:

Base on your comment, Figure.1g has been changed to Figure.2a in our revised manuscript.

- For Figure 2d, are the differences in mean beta values between CAR T and dCAR T at these gene loci statistically significant? Please indicate on the graph and/or in the figure legend.

Authors' response:

Thank you for mentioning this point. The significant different methylation of the CpG sites ($p < 0.01$) were shown in Fig. 2 between the dCAR T cells and CAR T cells. We apologize for not clearly describing this point. The significant value has been added to the Fig.2 legend in our revised manuscript.

- In Figure 3, authors argue that the superior proliferation and maintenance of a memory-like phenotype of dCAR T cells after repeated (3x) antigen exposure occurs via prevention of T cell exhaustion. While I agree that these particular observations are interesting and important, there is not enough evidence to claim that these benefits occur by mitigating exhaustion (at least, at this point in the paper). To claim this, authors would need to show that 1) CAR T cells exhibit a decline in anti-tumor functionality after 3x antigen stimulation and 2) decitabine treatment prevents this dysfunction. I would advise authors to change the title of the figure to something along the lines of, "dCAR T cells exhibit enhanced anti-tumor reactivity and maintenance of a memory-like phenotype at low effector:target ratios", but mention in the manuscript text that these data suggest that prevention of exhaustion could be one mechanism by which decitabine treatment enhances

anti-tumor
functionality.

Authors' response:

We appreciate your excellent suggestion. Our revisions in response to this comment have improved the manuscript. First, we changed the title of Fig. 3 to “**dCAR T cells exhibit enhanced antitumour reactivity and maintenance of a memory-like phenotype at low effector:target ratios**” based on your suggestion. Second, based on your above comment, we added a comparative experiment to verify the antitumour function of the CAR T and dCAR T cells after the time of tumour cell stimulation was prolonged. The results showed that compared to CAR T cells, dCAR T cells exhibited lower proportions of PD-1- and TIM3-positive cells and produced higher levels of TNF and IFN γ upon re-stimulation with Raji cells. These results “**These results were confirmed by further extension of the tumour cell stimulation time. After two weeks of constant stimulated by low-dose Raji cells, CAR T cells exhibited higher proportions of PD1+TIM3+ cells than dCAR T cells (Supplemental. Fig. 4a). Compared to the CAR T cells, the dCAR T cells produced higher levels of IL-2, TNF- α and IFN- γ upon re-stimulation with Raji cells (Supplemental Fig. 4b).**” were added to lines 126-131 on pages 6-7. Finally, in reference to your perceptive comment, we have changed the sentence, “Taken together, these results indicated that dCAR T cells retained robust effector functions and downregulated the expression of exhaustion-associated markers upon repeated antigen presentation.” to “**These results indicated that the dCAR T cells retained robust effector functions and exhibited upregulated expression of the memory-associated markers. The data obtained following repeated antigen presentation suggested that prevention of exhaustion could be one mechanism by which DAC treatment enhanced anti-tumour functionality.**” in the revised manuscript (lines 131-134, page 7).

• The figure legend for 3d does not match the figure (which is a heatmap). It reads, “Bars show mean values with data points for 3 independent experiments. Experiments were repeated with four donor-derived T cells (n = 4). A two-tailed, paired, two-sample t test was used for statistical analysis. * P < 0.05, **P < 0.01, ***P < 0.001.” Are these heatmaps mean values of 4 donors? Are there supposed to be bar graphs here instead of heatmaps?

Authors' response:

Thank you for noting this issue. The legend for Fig. 3d was inaccurate. The heat map contained data from 3 independent experiments, and the mean values are expressed by shade. We have changed the heat map showing these results to a bar graph representation in the revised manuscript.

- Please specify the difference between the two plots shown in Figure 4e. I assume the left plot was obtained from CD4s and the right from CD8s?

Authors' response:

The instructions for CD4 and CD8 T cells have been added to the **Fig. 4**.

- Figures 6f and 7i show histograms that are intended to demonstrate an increase in the percentage of CAR+ cells in the blood of mice treated with dCAR T cells. It would be nice if these data were quantified (either embedded in the histograms themselves) or represented in bar graph form.

Authors' response:

Thank you. Base your comment, the quantified data have embedded in the graph of **Fig. 6f** and **Fig. 8a**.

- Figure 9b – What statistical test was used here? Given the size of the error bars between the two groups being compared, I can't imagine there is statistical significance.

Authors' response:

In our previous manuscript, the unpaired t-test was used for statistical analysis of the sample data collected on the 0th, 7th, and 14th days shown in Fig. 9b. There was no difference in tumour size between the dCAR T and CAR T cell groups on days 0 and 7, but tumour size was significantly different between the two groups on the 14th day. To clarify the results, we marked the results of statistical analysis of the differences at each time point in **Fig.9b**, and added the following text regarding the statistical methods to the revised legend: **“Two-tailed, unpaired, two-sample t-tests of the data taken at each time point were used for statistical analyses. **p < 0.01.”**

- Figure 9e – Please change the Y-axis to include the specific markers used assess Tcm phenotype (CD62L+ CD45RO+).

Authors' response: These changes were made per your suggestion.

- Figure 10 – Please indicate how many samples were analyzed via RNA-seq in the figure legend.

Authors' response:

Thank you for your comment. We added the number of tumour-infiltrating T cell samples analysed to the figure legend and simultaneously supplemented the Materials and Methods section to indicate the collection method and quantity of tumour-infiltrating CAR T cell samples used in the experiment. The following text was added to the **Materials and Methods** section of the revised manuscript: **“Isolation of tumor infiltrating CAR T for subsequent analyses. On day 7 and 14 after cell infusion, 5-8 mice were killed and removal of tumour. Tumours were collected, pooled together by group, homogenized, and then dissociated using the MACS Miltenyi Mouse Tumour Dissociation kit (Miltenyi Biotec) according to manufacturer’s instructions. Purified CAR T cells were obtained by sorting (magnetic beads, Miltenyi Biotec) according to manufacturer’s instructions. For flow cytometry analyses, 500000 cells for each group were sorted from the isolated tumor infiltrating CAR T cells. For RNA-seq, two technical replicates of 10⁶ cells each were sorted from the isolated tumor infiltrating CAR T cells.”** (lines 367-374, pages 17-18)

- Figure 10a – Does this heatmap represent mean expression values from multiple samples for each condition/timepoint? Please specify.

Authors' response:

Based on your comment, the following text was added to the Figure legend: **“The overall expression of gene transcript profiles in the dCAR and CAR T cells on days 7 and 14 after cell treatment. Mean average plots of genes differentially expressed in tumor infiltrating dCAR T cells versus tumor infiltrating CAR T cells on different time point after cell infusion. P values calculated using Wald test (as implemented in DESeq2). Differentially expressed genes (Pvalue < 0.01, fold change (log₂ scale) ≥ 1 or ≤ -1) are highlighted.”** (lines 845-849, page 43).

At meanwhile, the text about RNA-seq analyses **“RNA-sequencing analysis. Prior to differential gene expression analysis, the read counts were adjusted for each sequenced library by edgeR program package through one scaling normalized factor. The DEseq2 package**

version 1.16.1 in the statistical analysis software R/Bioconductor was used to analyse the normalized RNA-seq data. Generalized linear models with empirical Bayes approach were used to identify significantly differentially expressed genes. The Benjamini-Hochberg (BH) method was used to adjust the raw P-values to control for the false discovery rate (FDR) in the data analysis. An adjusted P-value of 0.05 or less was regarded as significant difference. For clustering, we clustered different samples to see the correlation using hierarchical clustering distance method with the function of heatmap using silhouette coefficient to adapt the optimal classification with default parameter in R.” were added in revised manuscript. (lines 426-435, page 20)

- Figure 10b – Please use arrows to designate data points corresponding to the gene labels.

Authors’ response: This was done per your suggestion.

- Figure 10c – How were genes selected for each of these modules (memory, exhaustion/activation, proliferation, death/aging)? Was it biased or unbiased selection? Are there statistically significant differences?

Authors’ response: The selection of modular genes was based on the T cell-related genes in the corresponding modules in the GO and KEGG databases, and there was no bias in their selection. The results showed the quartile of all gene expression values in each module, and the analysis was not an analysis of differential expression. Based on your questions, to more clearly express the difference between CAR T cell and dCAR T cell genes involved in "memory, proliferation and exhaustion", we conducted GSEA analyses and have added the results to the Figure. The text, **“GSEA revealed upregulated the expression of memory-associated and cell proliferation-related GO items in the dCAR T cells compared with the CAR T cells on days 7 and 14 (Fig. 9i). Compared with tumour-infiltrating CAR T cells, the tumour-infiltrating dCAR T cells showed the upregulated expression of activation/ exhaustion-associate genes on day 7 and downregulated these genes on day 14 after cell infusion (Fig. 9j).”** was added to the revised manuscript. (lines 269-274, page 13)

- Subfigures B and C in Supplementary Figure 2 do not match their descriptions in the figure legend, please correct this.

Authors' response: We have corrected the sub-figures description in the revised manuscript.

- Please include the number of donors/samples for which bisulfite sequencing was conducted in Supplementary Figure 6.

Authors' response: The number of donors from whom samples for bisulfite sequencing were obtained has been added to the revised figure legend.

To Reviewer 2:

Dear Reviewer:

All authors express gratitude for your positive evaluation of our manuscript and your elaborate suggestions for revisions. Your comments were very instructive and valuable for revising our manuscript. We agree with your comments, and we have carefully revised our manuscript as detailed in the following point-by-point responses.

Your comments:

· In the new figure 9f, the authors showed the frequency of CD25+ / CD3+ T cells is higher in dCAR T without differentiating CD4+ from CD8+ in this analysis. Activation of these cells could be more accurately shown by using activation markers such as CD44 and CD69, in addition to CD25 in this in vivo experiment. Could the authors analyse on the RNAseq data of this experiment to support the activation state of CD4+ and CD8+ tumour infiltrating CAR and dCAR T.

Authors' response:

Thank you very much for your meticulous review. In our future research, we will add more cell surface indicators to define the T cell activation state. Due to the very limited number of sorted CAR T cells in tumours, we could not sort CD4- and CD8-positive tumour-infiltrating CAR T cells separately for RNA-seq analysis. Based on your suggestion, we further analysed the differences in activation-associated gene expression between the tumour-infiltrating CAR T cells and dCAR T cells. The dCAR T cells showed the upregulation of activation- and cytokine production-related genes on both the 7th and 14th days. The results of GO and GSEA analyses showed that the tumour-infiltrating dCAR T cells were significantly enriched in genes involved in the T cell activation compared with CAR T cells. This result has been added to the revised manuscript. The text, **“The tumour infiltrating dCAR T cells significantly upregulated the expression of cytokine- and T cell activation-related genes relative to that of the tumour-infiltrating CAR T cells on the 7th day and 14th days (Fig. 9j and Supplemental Fig.10).”** to the revised manuscripts. (lines 279-285, pages 13-14)

· As mentioned in the text that CD8 dCAR T cells showed stronger expansion and viability than CAR T cells (line 246-247). However, a mixed CD4+/CD8+ result was presented in the figure

(figure 9), which doesn't support the statement in the main text.

Authors' response:

Thank you for noting this issue. We have deleted "CD8-positive" to correct the description in the revised manuscript.

· The authors used a lot of "exhaustion markers" measuring exhaustion phenotype for both continuous/repetitive antigen stimulations (figure 3g-i) and single tumour inoculation (figure 9). The markers, such as PD-1, can be up-regulated upon activation without the cells being "exhausted". As such, while agreeing with that dCAR T cells can be less exhausted under repetitive antigen stimulation, the evidence doesn't support that tumour-infiltrating dCAR T cells are less "exhausted" in a short-term single tumour inoculation. Could the authors show or discuss on cytokine expression/secretion with RNAseq results to support the interpretation.

Authors' response:

We appreciate your perceptive comments, which expanded our knowledge of T cell exhaustion. Based on your comments and those of another review, first, we performed additional experiments to compare the cytokine release functions of dCAR T and CAR T cells after repeated antigen stimulation for an extended period of time in vitro. The results showed that two weeks after constant stimulation by CD19-positive tumour cells, dCAR T cells produced higher levels of TNF- α and IFN- γ upon re-stimulation with Raji cells than the CAR T cells did. The results **“These results were confirmed by further extension of the tumour cell stimulation time. After two weeks of constant stimulation by low-dose Raji cells, CAR T cells exhibited higher proportions of PD1+TIM3+ cells than dCAR T cells (Supplemental Fig. 4a). Compared to the CAR T cells, the dCAR T cells produced higher levels of IL-2, TNF- α and IFN- γ upon re-stimulation with Raji cells (Supplemental Fig. 4b). These results indicated that the dCAR T cells retained robust effector functions and exhibited upregulated expression of the memory-associated markers. The data obtained following repeated antigen presentation suggested that prevention of exhaustion could be one mechanism by which DAC treatment enhanced anti-tumour functionality.”** was added to the revised manuscript (lines 126-133, pages 6-7).

Second, combined your comment above, we added the results of an analysis of differences in cytokine-related gene expression between the tumour-infiltrating dCAR T cells and CAR T cells on

days 7 and 14. The results showed that the tumour-infiltrating dCAR T cells exhibited significantly increased expression of T cell cytokine and T cell activation related genes relative to the tumour-infiltrating CAR T cells on the 7th day and 14th day. In addition, the tumour-infiltrating dCAR T cells on day 14 showed no significant difference in cytokine-related gene expression compared with the tumour-infiltrating dCAR T cells on day 7, while the tumour-infiltrating CAR T cells on day 14 exhibited decreased expression of cytokine-related genes compared to their levels in CAR T cells on day 7. Combined with the results of cytokine detection and RNA-seq data, these findings show that after 14 days of cell infusion, the cytokine function of the CAR T cells was reduced; in contrast, the tumour-infiltrating dCAR T cells showed no decrease in cytokine function. These data suggest that the tumour-infiltrating CAR T cells were in an exhausted/dysfunction state relative to the tumour-infiltrating dCAR T cells on day 14 after cell infusion. The sentences, **“GESA revealed upregulated the expression of memory-associated and cell proliferation-related GO items in the dCAR T cells compared with the CAR T cells on days 7 and 14 (Fig. 9i). Compared with tumour-infiltrating CAR T cells, the tumour-infiltrating dCAR T cells showed the upregulated expression of activation/ exhaustion-associate genes on day 7 and downregulated these genes on day 14 after cell infusion (Fig. 9j). Throughout, the tumour-infiltrating dCAR T cells had higher levels of memory-associated genes (Supplemental Fig. 10); and the dCAR T cell group also exhibited stronger expansion and viability of the tumour-infiltrating dCAR T cells, which showed increased population of Tcm and CD25-positive cells, than those of the CAR T cell group (Fig.9.e and f); thus, the relatively less exhausted status of the dCAR T cells in the later period may be because a greater proportion of these cells were stem cell-like cells. The tumour infiltrating dCAR T cells significantly upregulated the expression of cytokine- and T cell activation-related genes relative to that of the tumour-infiltrating CAR T cells on the 7th day and 14th days (Fig. 9j and Supplemental Fig.10). Combined with cytokine detection (Fig. 8c and d) and the RNA-seq data, the cytokine produced function was reduced in the CAR T cells after 14 days of cell infusion; in contrast, the tumour-infiltrating dCAR T cells showed stable in cytokine produced function. These data suggested that the tumour-infiltrating dCAR T cells were in a relatively dysfunction state relative to the tumour-infiltrating dCAR T cells on day after cell infusion.”** to the revised manuscripts. (lines 269-285, pages 13-14)

· Fig 1d: the authors use the term ‘resting CAR T cells’. Are these cells resting or untreated with DAC?

Authors’ response: Thank you for your comprehensive review. The “resting CAR T cells” in this study represent CAR T cells that have not been stimulated by antigens or tumour cells. Based on your comment, we added detail describing the “resting state” where this phrase first appears in the manuscript. (line 67, page 4)

· Fig 1f: how is the expression of CAR verified?

Authors’ response:

In this study, we used CAR T cells purified by magnetic bead sorting for various experiments, including in vitro experiments, western blotting, animal experiments and omics analysis. While performing western-blot detection of DNMT3a, we also tested the expression of CAR gene use CD3z protein. There was no difference in CD3z expression between dCAR T and CAR T. We add this result and methods to the revised manuscript. To describe this data more clearly, we added the sentence **“We found that low-dose, short-term DAC treatment in vitro persistently induced the degradation of DNMT3a (Fig. 1f) but did not affect the expression of CAR (data not shown).” (lines 69-71, page 4)** to **Result** section, and added the sentences **“The purified CAR T and dCAR T cells were obtained by magnetic bead sorting, purity was determined by flow cytometry.”**and **“The expression of CAR gene use CD3z protein to determine. The total protein was separated on a 4-12% SDS-PAGE gel followed by standard immunoblotting with antibody to β -actin (Abcam), antibody to DNMT3a (Abcam) and antibody to CD3zeta (Abcam).”** to **Materials and Methods** section, in revised manuscript. (lines 376-381, page 18)

· Fig 2: The figures require higher resolution as it is not possible to read any of the legends.

Authors’ response:

Based on your suggestion, we replaced the figure with a higher resolution figure in our revised manuscript (Fig. 2). Please let us know if anything affects your ability to read the picture, and we will further increase the resolution of the picture.

- Fig 3i: Correct EOMES.

Authors' response: Thank you for noting this spelling mistake, which has been corrected in the revised manuscript.

- Fig 4c: Raji model was used but the experiment set up was not mentioned properly in the text.

Authors' response: Thank you very much for your meticulous review. We apologize for the incorrect description in the Fig. 4 legend. Due to the misordering of Fig. 4c and d, the legend did not match the panel. We have carefully checked the order of the legend and panels in the revised manuscript and corrected this incorrect description. The sentences “**d, The proliferation fold change was obtained by the number of cell counts on the 10th day after cell culture compared with the number of cells before cell culture. e, Cell count analysis of CAR T cell proliferation as measured after 24 h of coculture with Raji cells at an E:T ratio of 1:1. Dots show mean values with data points for 5 independent experiments.**” to the revised manuscript. (lines 704-707, page 33)

- Fig 4d: It is described in the text that both CD4 and CD8 positive dCAR T cells show higher proliferation than CAR T cells. This statement is not valid for CD8+ cells in figure 4d.

Authors' response:

Another reviewer also made the same comment. There was some ambiguity in the text in our previous manuscript. The text “Although the CD8 dCAR T cells proliferation was lower than those of the CAR T cell group in ex vivo, this relatively lower proliferation rate was reversed after subsequent antigen stimulation in vitro. For both the CD4 and CD8 T cells, the amplification capacity was significantly higher in the dCAR T cell group than those of in the CAR T group under repeated antigen stimulation (Fig.4.d).” were revised to “**Furthermore, the amplification capacity of both the CD4-positive and CD8-positive T cells was remarkably higher in the dCAR T cell group than that of in the CAR T cell group under repeated antigen stimulation (Fig. 4e).**” (lines 141-143, page 7).

- Fig 2c is inappropriately referred in line 105

Authors' response:

Thanks to your instructive notation, the reference to Fig. 2c has been modified in the revised manuscript.

· Could the authors ensure that the figures are referred correctly in the main text, as well as a more readable and correct sentence structure in the manuscript.

Authors' response:

Thank you very much for your meticulous review. We acknowledge that the language was not always correct and that some figure panels were not correctly referenced in the previous version of the manuscript. To address these important issues, we have asked native English-speaking friends to edit the entire manuscript and carefully checked the reference for each figure panel in the manuscript to ensure that these references are correct.

To Reviewer 3:

Dear Reviewer:

All authors greatly appreciate your positive evaluation of our manuscript. The comments you made were very instructive and helpful for revising our manuscript. We agree with your comments, and we have revised our manuscript as detailed below in our point-by-point responses.

Your comments:

The pre-processing of RNA-seq data (such as the sequence alignment) seems missed in the revised version of the manuscript.

Authors' response:

Thank you very much for your meticulous review. Based on your comment, we added **“RNA sequencing data processing. Raw data of fastq format were firstly processed, clean data (clean reads) were obtained by removing reads containing adapter, reads containing ploy-N and low-quality reads from raw data. The quality control of raw fastq data was by FastQC. Index of the reference genome was built using Hisat2 v2.0.5 and paired-end clean reads were aligned to the reference genome using Hisat2 v2.0.5. Pair-end reads were trimmed using Trim Galore (<https://github.com/FelixKrueger/TrimGalore>). Cleaned reads were mapped to UCSC Human GRCh38/hg38 (<http://genome.ucsc.edu/>). The mapped reads of each sample were assembled by StringTie (v1.3.3b) .Trimmed reads with a mapping quality exceeding 20 were counted by HTSeq (Hisat2 v2.0.5).FPKM of each gene was calculated based on the length of the gene and reads count mapped to this gene.”** to the revised manuscripts. (lines 415-424, pages 19-20)

- The RNA-seq analysis description need to specify the normalization method they used in the data analysis.

Authors' response:

Thank you very much for your meticulous review. Based on your suggestion, we added five sections, including the **“Isolation of tumor infiltrating CAR T for RNA-seq analyses”, “RNA sequencing data processing”, “RNA-sequencing analysis.”, “GO enrichment analysis of differentially expressed genes”** and **“Gene set enrichment analysis (GSEA)”** to Materials and

Methods of revised manuscript.

Isolation of tumor infiltrating CAR T for subsequent analyses. (line 366-373, page17)

On day 7 and 14 after cell infusion, 5-8 mice were killed and removal of tumour. Tumours were collected, pooled together by group, homogenized, and then dissociated using the MACS Miltenyi Mouse Tumour Dissociation kit (Miltenyi Biotec) according to manufacturer's instructions. Purified CAR T cells were obtained by sorting (magnetic beads, Miltenyi Biotec) according to manufacturer's instructions. For flow cytometry analyses, 500,000 cells for each group were sorted from the isolated tumor infiltrating CAR T cells. For RNA-seq, two technical replicates of 10^6 cells each were sorted from the isolated tumor infiltrating CAR T cells.

RNA sequencing data processing. (line 415-424, page20)

Raw data of fastq format were firstly processed, clean data (clean reads) were obtained by removing reads containing adapter, reads containing ploy-N and low quality reads from raw data. The quality control of raw fastq data was by FastQC. Index of the reference genome was built using Hisat2 v2.0.5 and paired-end clean reads were aligned to the reference genome using Hisat2 v2.0.5. Pair-end reads were trimmed using Trim Galore (<https://github.com/FelixKrueger/TrimGalore>). Cleaned reads were mapped to UCSC Human GRCh38/hg38 (<http://genome.ucsc.edu/>). The mapped reads of each sample were assembled by StringTie (v1.3.3b). Trimmed reads with a mapping quality exceeding 20 were counted by HTSeq (Hisat2 v2.0.5). FPKM of each gene was calculated based on the length of the gene and reads count mapped to this gene.

RNA-sequencing analysis. (line 425-434, page20)

Prior to differential gene expression analysis, the read counts were adjusted for each sequenced library by edgeR program package through one scaling normalized factor. The DEseq2 package version 1.16.1 in the statistical analysis software R/Bioconductor was used to analyze the normalized RNA-seq data. Generalized linear models with empirical Bayes approach were used to identify significantly differentially expressed genes. The Benjamini-Hochberg (BH) method was used to adjust the raw P-values to control for the false discovery rate (FDR) in the data analysis. An adjusted P-value of 0.05 or less was regarded as significant difference. For clustering, we clustered different samples to see the correlation using hierarchical clustering distance method with the function of heatmap using silhouette coefficient to adapt the optimal classification with default parameter in R.

GO enrichment analysis of differentially expressed genes. (line 435-439, page20)

Gene Ontology (GO) enrichment analysis of differentially expressed genes was implemented by the cluster Profiler R package, in which gene length bias was corrected. GO terms with corrected P value less than 0.05 (or P value less than 0.05) were considered significantly enriched by differential expressed genes.

Gene set enrichment analysis (GSEA). (line 440-445, page21)

Raw data were normalized in two replicates for GSEA analysis. The genes were ranked according to the degree of differential expression in the two samples, and then the predefined Gene Set were tested to see if they were enriched at the top or bottom of the list. We use the local version of the GSEA analysis tool <http://www.broadinstitute.org/gsea/index.jsp>, GO, KEGG, Reactome, DO, DisGeNET data set were used for GSEA independently.

- The version of the DESeq2 package need to be specified.

Authors' response:

Thank you for noting this issue. DESeq2 package version 1.16.1 was used in RNA-seq analyses, the version number was added to the revised manuscript.

“Then calculate the hypothesis test probability (p-value) through the statistical model, and finally perform multiple, repeated hypothesis test correction (BH) to get the FDR value (false discovery rate, padj is its common form, the following are all used padj to represent FDR) . DESeq2 software was used to analyse the significance of expression difference, and padj was less than 0.05 as the difference significance standard.” Flows better if revised to “The DEseq2 package version 1.16.1 in the statistical analysis software R/Bioconductor was used to analyze the normalized RNA-seq data. Generalized linear models with empirical Bayes approach were used to identify significantly differentially expressed genes. The Benjamini-Hochberg (BH) method was used to adjust the raw P-values to control for the false discovery rate (FDR) in the data analysis. An adjusted P-value of 0.05 or less was regarded as significant difference.”

Authors' response:

We appreciate your excellent suggestion. Base on your comments, the sentences “Then calculate the hypothesis test probability (p-value) through the statistical model, and finally perform

multiple, repeated hypothesis test correction (BH) to get the FDR value (false discovery rate, padj is its common form, the following are all used padj to represent FDR) . DESeq2 software was used to analyse the significance of expression difference, and padj was less than 0.05 as the difference significance standard.” were revised to **“The DEseq2 package version 1.16.1 in the statistical analysis software R/Bioconductor was used to analyze the normalized RNA-seq data. Generalized linear models with empirical Bayes approach were used to identify significantly differentially expressed genes. The Benjamini-Hochberg (BH) method was used to adjust the raw P-values to control for the false discovery rate (FDR) in the data analysis. An adjusted P-value of 0.05 or less was regarded as significant difference.”** (line 424-432, page 20)

REVIEWERS' COMMENTS

Reviewer #2 (Remarks to the Author):

The revised manuscript 'Low-dose decitabine priming endows CAR T cells with enhanced and persistent anti-tumour potential via epigenetic reprogramming' is now substantially improved and a number of my prior concerns have been addressed.

Some over-generalisations and typographic errors still remain throughout the manuscript and these should again be examined by a native language speaker or editorial team within Nature Communications prior to publication.

Minor corrections:

Line 59: The phrase proliferative 'capacity' needs to be completed.^[1]Line 146: The characterization 'remarkably' is an overstatement based on the data of figure 4e.^[1]Line 268: The word 'positive' is misspelled.

Figure 6c: Blue colour should be exchanged with red for dCAR in the labels.

Reviewer #3 (Remarks to the Author):

The authors have addressed all my comments. No additional concerns.

Dear Reviewer:

All authors express great thanks for your positive evaluation of our manuscript. Your comments are very helpful for us. I would like to fully accept your comments and have carefully revised our manuscript as follows. According to your comments, I have given a point-by-point revised as follows.

Your comments:

Some over-generalisations and typographic errors still remain throughout the manuscript and these should again be examined by a native language speaker or editorial team within Nature Communications prior to publication.

Authors' response:

Thank you very much for your meticulous review. Based on your comments and those of editorial, we removed exaggerated languages from revised manuscript, and used Nature Research Editing Service to edit entire manuscript.

Minor corrections:

Line 59: The phrase proliferative 'capacity' needs to be completed. Line 146: The characterization 'remarkably' is an overstatement based on the data of figure 4e. Line 268: The word 'positive' is misspelled.

Figure 6c: Blue colour should be exchanged with red for dCAR in the labels.

Authors' response:

Thank you for noting these mistakes, which have been corrected in the revised manuscript.